

Solid Earth

# A reconstruction of Iberia accounting for Western Tethys–North Atlantic kinematics since the late-Permian–Triassic

**Paul Angrand[1], Frédéric Mouthereau[1], Emmanuel Masini[2,3], and Riccardo Asti[4]**

[1]Geosciences Environnement Toulouse (GET), Université de Toulouse, UPS, Univ. Paul Sabatier, CNRS,
IRD, 14 av. Edouard Belin, 31400 Toulouse, France CE1
[2]M&U sas, Saint-Égrève, France CE2
[3]ISTerre, Université Grenoble Alpes, France TS1
[4]Université de Rennes, CNRS, Géosciences Rennes-UMR 6118, 35000 Rennes, France

**Correspondence:** Paul Angrand (paul.angrand@get.omp.eu)

**Abstract.** CE3 The western European kinematic evolution results from the opening of the western Neotethys and the Atlantic oceans since the late Paleozoic and the Mesozoic. Geological evidence shows that the Iberian domain recorded the propagation of these two oceanic systems well and is therefore a key to significantly advancing our understanding of the regional plate reconstructions. The late-Permian–Triassic Iberian rift basins have accommodated extension, but this tectonic stage is often neglected in most plate kinematic models, leading to the overestimation of the movements between Iberia and Europe during the subsequent Mesozoic (Early Cretaceous) rift phase. By compiling existing seismic profiles and geological constraints along the North Atlantic margins, including well data over Iberia, as well as recently published kinematic and paleogeographic reconstructions, we propose a coherent kinematic model of Iberia that accounts for both the Neotethyan and Atlantic evolutions. Our model shows that the Europe–Iberia plate boundary was a domain of distributed and oblique extension made of two rift systems in the Pyrenees and in the Iberian intra-continental basins. It differs from standard models that consider left-lateral strike-slip movement localized only in the northern Pyrenees in introducing a significant strike-slip movement south of the Ebro block. At a larger scale it emphasizes the role played by the late-Permian–Triassic rift and magmatism, as well as strike-slip faulting in the evolution of the western Neotethys Ocean and their control on the development of the Atlantic rift.

## 1 Introduction

Global plate tectonic reconstructions are mostly based on the knowledge and reliability of magnetic anomalies that record age, rate, and direction of sea-floor spreading (Stampfli and Borel, 2002; Müller et al., 2008; Seton et al., 2012). Where these constraints are lacking or their recognition is ambiguous, kinematic reconstructions rely on the description and interpretation of the structural, sedimentary, igneous and metamorphic rocks of rifted margins and orogens (e.g., Handy et al., 2010; McQuarrie and Van Hinsbergen, 2013). However, the required quantification and distribution of finite strain into deformed continents often remain uncertain due to the poor preservation of pre-kinematic markers.

A well-known example of this problem is illustrated by the contrasting Mesozoic plate kinematic models proposed for the Iberian plate relative to Europe with significant implications for the reconstructions of the Alpine Tethys and Atlantic oceans (Olivet, 1996; Handy et al., 2010; Sibuet et al., 2004; Vissers and Meijer, 2012; Barnett-Moore et al., 2016; Nirrengarten et al., 2018). This movement is proposed to be imposed by the northward propagation of the North Atlantic rifting during the Triassic–Early Cretaceous period (Olivet, 1996; Stampfli et al., 2001; Handy et al., 2010; Barnett-Moore et al., 2016; Nirrengarten et al., 2018). Although all of the proposed reconstructions agree on the amplitude and kinematics of a 400–500 km left-lateral strike-slip motion between Europe and Iberia after the Variscan orogeny, its precise timing and spatial partitioning is debated and remains unresolved so far (e.g., Vissers and Meijer, 2012; Barnett-

Moore et al., 2016). Because of the lack of geological constraints, such a large strike-slip displacement has been supposedly exported along the North Pyrenean Fault (Fig. 1a) (e.g., Choukroune and Mattauer, 1978; Debroas, 1987, 1990; Lagabrielle et al., 2010; Jammes et al., 2009). Here again, previous studies were not conclusive so there are currently no firm geological constraints nor geophysical evidence across the Pyrenees to argue for significant transcurrent deformation during the Jurassic or the Cretaceous (Olivet, 1996; Masini et al., 2014; Canérot, 2016; Chevrot et al., 2018). Other studies have suggested that this conclusion might hold true back to the Permian, based on local geological evidence from the western Pyrenees (Saspiturry et al., 2019).

An alternative scenario has recently emerged (Tugend et al., 2015; Nirrengarten et al., 2018; Tavani et al., 2018), proposing a spatiotemporal partitioning of the deformation in a wider deformation corridor than the single Pyrenean belt. It suggests that the transcurrent deformation that results from the eastwards movement of Iberia occurred mainly during the Late Jurassic–Early Cretaceous in northern Iberia along a 100 km scale pull-part or en échelon rift basins formed by the NW–SE-trending Iberian massifs. Indeed, along these massifs, several extensional basins recorded major subsidence and strike-slip deformation during the late-Permian to middle-Cretaceous time interval (Alvaro et al., 1979; Salas et al., 2001; Aldega et al., 2019; Aurell et al., 2019; Soto et al., 2019). However, no geological evidence for lithosphere-scale strike-slip movements has yet been clearly defined in the intra-Iberian basins.

These basins separate the Ebro continental block from the greater Iberia to the south. Ebro CE4 is delimited to the north by the Pyrenean system. Extension then migrated and localized to the north (Rat et al., 2019) leading to oceanic spreading in the Bay of Biscay and hyper-extension in the Pyrenean rift basins (Jammes et al., 2009; Lagabrielle et al., 2010; Mouthereau et al., 2014; Tugend et al., 2014, 2015).

Moreover, the contribution of pre-Late Jurassic–Early Cretaceous extension phases might have been substantial to the overall crustal attenuation and movements of Iberia (Fig. 1a) (Fernández, 2019; Soto et al., 2019). Indeed, two major geodynamic events, the late-Permian–Early Triassic breakup of Pangea and the opening of the Neotethys and the Late Triassic–Early Jurassic central Atlantic magmatic event preceding the opening of the North Atlantic Ocean are recorded in Iberia and contributed to the finite crustal thinning. Therefore, all full-fit reconstructions considering that extension between Iberia and Newfoundland only initiated by Jurassic times in the North Atlantic realm invariably overestimate the amount of strike-slip motion required in the Pyrenees and northern Iberia from the Jurassic onward (Barnett-Moore et al., 2016; Nirrengarten et al., 2018).

Here, we examine the possible contribution of the late-Permian–Triassic extension to the plate reconstruction of Iberia between the Neotethys and the North–central Atlantic domains and its impact on the definition of the spatial and temporal distribution of strike-slip movement between Iberia and Europe. By integrating constraints from 270 to 100 Ma, our reconstructions bring to light the connection between the Tethyan and the Atlantic oceanic domains.

## 2   Late-Permian–Triassic rifting and magmatism in the North Atlantic and western Europe

The tectonic and thermal evolution of the "Iberian buffer" between Africa and Europe at the Permian–Triassic boundary reflects the complex post-Variscan evolution of the Iberian lithosphere. This domain has in fact experienced significant Permian crustal thinning in relation to the post-orogenic collapse of the Variscan belt (De Saint Blanquat et al., 1990; de Saint Blanquat, 1993; Vissers, 1992; Saspiturry et al., 2019) and the fragmentation of the Gondwana margin more broadly (Schettino and Turco, 2011; Stampfli and Borel, 2002; Ziegler, 1989, 1990). This late-Permian–Lower Triassic phase is associated with the deposition of thick detrital non-marine deposits in intra-continental basins. Sedimentation became carbonaceous during the Middle Triassic. Finally, the Late Triassic is characterized by a thick evaporitic (mainly salt) sequence (e.g., Ortí et al., 2017). However, this phase also resulted in lithospheric mantle delamination and thinning (Malavieille et al., 1990; Fabriès et al., 1991, 1998; Ziegler et al., 2004; Ziegler and Dèzes, 2006; Denèle et al., 2014).

Crustal thinning, attested by thick late-Permian–Triassic detrital rift basins deposited above an erosive surface, is well documented on seismic lines along the Atlantic margins (Fig. 2): the Nova Scotia–Moroccan basins (Welsink et al., 1989; Deptuck and Kendell, 2017; Hafid, 2000); Iberia–Grand Banks (Balkwill and Legall, 1989; Leleu et al., 2016; Spooner et al., 2019); the southern North Atlantic (Tankard and Welsink, 1987; Doré, 1991; Doré et al., 1999; Štolfová and Shannon, 2009; Peace et al., 2019b; Sandoval et al., 2019); the North Western Approaches CE5 (Avedik, 1975; Evans, 1990; McKie, 2017); and the North Sea (McKie, 2017; Jackson et al., 2019; Hassaan et al., 2020; Phillips et al., 2019). Onshore Iberia (Arche and López Gómez, 1996; Soto et al., 2019) and in the Pyrenean–Provence domains (Lucas, 1985; Espurt et al., 2019; Cámara and Flinch, 2017; Bestani et al., 2016) (Fig. 1b), an angular unconformity is observed between the Paleozoic and the Permian–Triassic strata (Fig. 2).

The Permian tectonic phase is contemporaneous with widespread magmatism related to the breakup of Pangea, and its transition toward diffuse extension. This is observed in present-day rifted margins of the North Atlantic such as the North Sea and Norwegian–Danish basins (Glennie et al., 2003), the Western Approaches (McKie, 2017), the Scottish Midland Valley (Upton et al., 2004), and the basement of Cenozoic collision belts around Iberia, for instance, in the Pyrenees (Lago et al., 2004a; Denèle et al., 2012; Vacherat

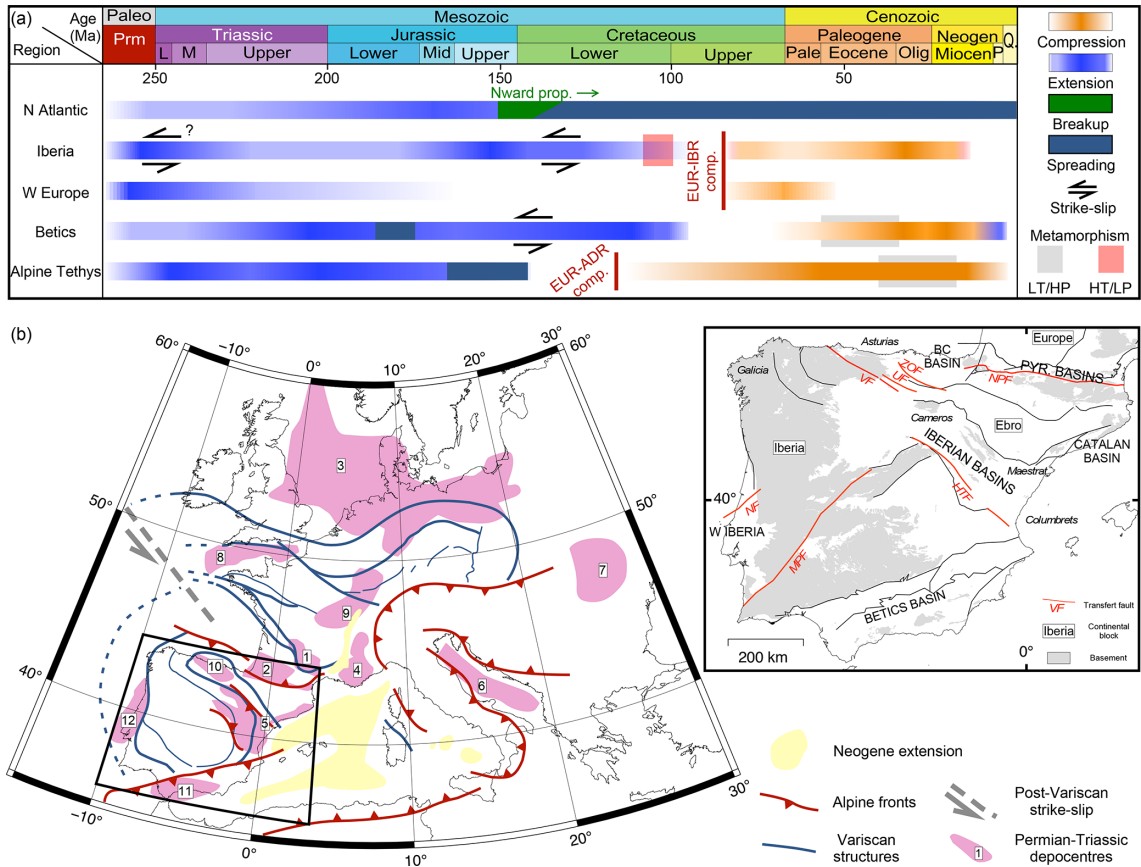

**Figure 1.** TS2 Geodynamic chart and localization of the study area. **(a)** Geodynamic chart of the main structural areas of the Iberian domain. BCB: Basque–Cantabrian Basin. **(b)** Location map of western Europe, showing the areas that are deformed in compression and extension and Permian–Triassic depocenters. (1) French Massif Central; (2) Aquitaine Basin and Pyrenees; (3) Germanic Basin; (4) southeastern France; (5) Iberian Basin; (6) Italy; (7) central Europe; (8) English Channel; (9) northeastern France; (10) Basque–Cantabrian Basin; (11) Betics; (12) West Iberia. Inset: Map of Iberia showing the main structures and transforms (red) and sedimentary basins (capitalized and italicized text) and sub-basins (italicized text). BC: Basque–Cantabrian; HTF: High Tagus Fault; MPF: Messejana–Plasencia Fault; NF: Nazaré Fault; NPF: North Pyrenean Fault; PYR: Pyrenean; UF: Ubierna Fault; VF: Ventaniella Fault; ZOF: Zamanza–Oña Fault.

et al., 2017; Saspiturry et al., 2019), the Iberian Range (Lago et al., 2004b), the Catalan Coastal Ranges (Solé et al., 2002), and the Betic Cordillera (Sánchez-Navas et al., 2017).

An expression of the continued lithospheric thinning and thermal instability associated with high heat flow during the Permian (McKenzie et al., 2015) and the Triassic (Peace et al., 2019a, and references therein). Lithospheric extension prior to (or associated with the premises of the subsequent) Early Jurassic continental breakup in the central Atlantic then favored the drainage of mantle melt reservoir (Silver et al., 2006; Peace et al., 2019a), attested by the very rapid emergence of the widespread tholeiitic magmatic CAMP (Central Atlantic Magmatic Province) event at the Triassic–Jurassic boundary (200 Ma) in the central Atlantic (Olsen, 1997; Marzoli et al., 1999; McHone, 2000). The CAMP extends to Iberia as large-scale volcanic intrusions such as the Messejana–Plasencia dyke (Cerbiá et al., 2003) in Iberia and the Late Triassic–Early Jurassic ophitic magmatism in the

Pyrenees (e.g., Azambre et al., 1987). Extension and salt movements in the North Sea basins during the Late Triassic further point to the propagation of the North Atlantic rift (Goldsmith et al., 2003).

The persistence of shallow-marine to non-marine deposition during this period contrasts with the large accommodation space that is required at a larger scale to sediment the giant evaporitic province in the late Permian (Jackson et al., 2019) and in the Late Triassic (Štolfová and Shannon, 2009; Leleu et al., 2016; Ortí et al., 2017). Therefore the subsidence appears much lower than that predicted by a simple isostatic model of crustal thinning (McKenzie, 1978).

Two hypotheses may be invoked to explain the difference with the McKenzie model. (1) The first is a reduction in mantle density during lithospheric thinning, due to mantle phase transitions to lighter mineral phases because of crustal attenuation (Simon and Podladchikov, 2008) and/or due to the trapping of melt in the rising asthenosphere before

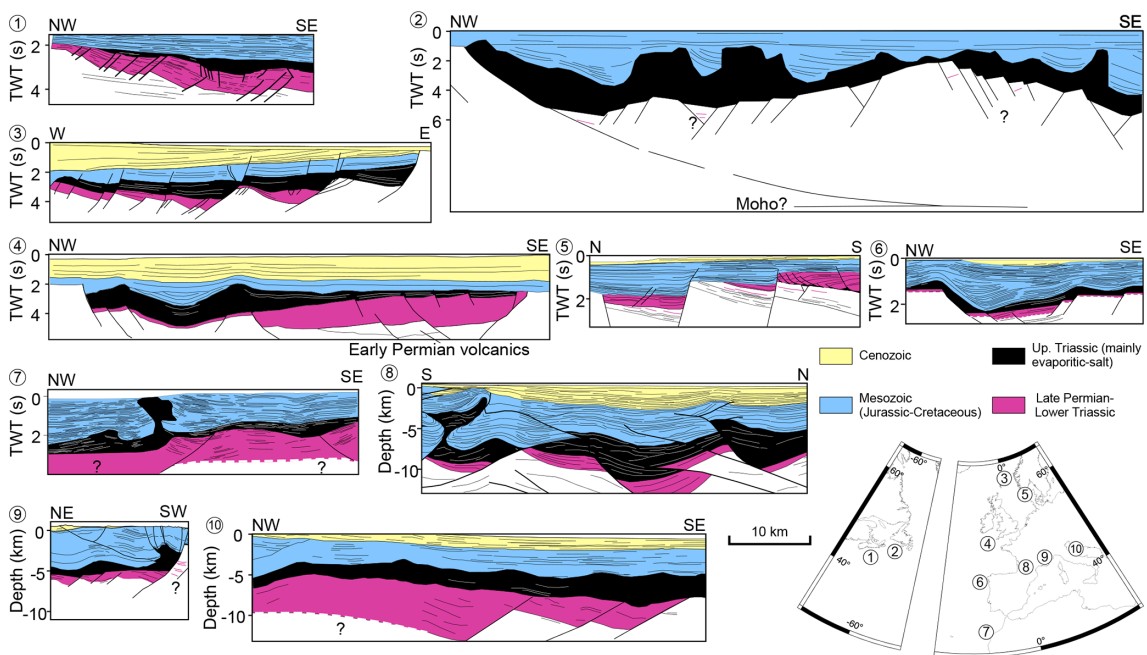

**Figure 2.** Compilation of interpreted seismic profiles along the North Atlantic margins and in western Europe. (1) Deptuck and Kendell (2017); (2) Balkwill and Legall (1989); (3)–(4) McKie (2017); (5) Philipps et al. (2019); (6) Rasmussen et al. (1996) TS3; (7) Hafid (2000); (8) Espurt et al. (2019); (9) Bestani et al. (2016); (10) Scisciani and Esestime (2017).

breakup (Quirk and Rüpke, 2018) in addition to magmatic re-thickening of attenuated crust by underplating. (2) Another possible hypothesis for the Permian–Triassic topographic evolution of the Iberian basins relies on the complex post-Variscan evolution of the Iberian lithosphere. Recent studies have shown that during the existence of the Pangea super-continent (300 to 200 Ma), temperature in the asthenospheric mantle increased due to the thermal insulation by the continental lid (Coltice et al., 2009; Ganne et al., 2016). This thermal insulation would be responsible for the accumulation of magmatic material of the CAMP (see Peace et al., 2019a, and references therein). Such a mantle thermal anomaly could have further inhibited lithospheric mantle re-equilibration after late-Variscan mantle delamination over a long time span. This model requires a strong impermeability of the overlying lithosphere (Silver et al., 2006). As a consequence of the Pangea breakup and magmatic emission at the Triassic–Jurassic boundary, the lithospheric mantle started to cool and thicken, causing isostatic subsidence of the thinned Iberian crust and resulting in topographic drop.

CE6 A first hypothesis to explain the difference with this model is that crustal attenuation induced density reduction in the thinned lithosphere by mantle phase transitions to lighter mineral phases during lithosphere thinning (Simon and Podladchikov, 2008) or due to the trapping of melt in the rising asthenosphere before breakup (Quirk and Rüpke, 2018) in addition to magmatic re-thickening of attenuated crust by underplating. Another possible hypothesis for the Permian–Triassic topographic evolution of the Iberian basins relies

on the complex post-Variscan evolution of the Iberian lithosphere. Recent studies have shown that during the existence of the Pangea supercontinent ($\sim 300$ to $\sim 200$ Ma), temperature in the asthenospheric mantle increased due to the thermal insulation by the continental lid (Coltice et al., 2009; Ganne et al., 2016). Such a mantle thermal anomaly could have further inhibited lithospheric mantle re-equilibration after late-Variscan mantle delamination over a long time span. Once mantle temperature dropped as a consequence of the Pangea breakup and magmatic emission at the Triassic–Jurassic boundary, the lithospheric mantle started to cool and thicken, causing isostatic subsidence of the thinned Iberian crust and resulting in topographic drop.

This argues for a protracted period of $\sim 100$ Myr (late Carboniferous to Late Triassic) of continental lithosphere thinning and magmatism prior to Early Cretaceous breakup of the North Atlantic but contemporaneous with the Tethyan evolution. One main consequence is that the late-Permian–Triassic extension has been so far underestimated in plate reconstructions, despite evidence for continuous extension.

## 3 From late-Permian–Early Triassic rifting to Late Jurassic-Early Cretaceous rifting in Iberia

The Permian–Triassic basins of Iberia are exposed in the inverted Mesozoic rift basins of the Basque–Cantabrian and Pyrenean belts, the Iberian Ranges, the Catalan Range and the Betic Cordillera (Alvaro et al., 1979; Lagabrielle et al.,

2020) (Figs. 1b and 3a). The coincidence between the orientations of the Alpine orogenic segments and the spatial distribution of Permian–Triassic depocenters (Figs. 1b and 3a) suggest that the Cenozoic orogenic cycle largely inherits the earliest stages of the Tethyan rift evolution. In addition, these Permian–Triassic depocenters are superposed over Variscan structures (Fig. 1b), suggesting antecedent tectonic control of the Tethyan continental rift segment by the late-Variscan evolution.

We analyze subsidence reconstructed based on a compilation of well data and a synthetic stratigraphic section in the Aquitaine Basin (Brunet, 1984), the Cameros and Iberian basins (Salas and Casas, 1993; Salas et al., 2001; Omodeo-Salé et al., 2017), western Iberia (Spooner et al., 2019), and the Betics (Hanne et al., 2003) to estimate 1D mean tectonic subsidence evolution in these areas (Fig. 3b; see Supplement for individual tectonic subsidence curves in each region). For each region, we calculated the mean tectonic subsidence, following the approach of Spooner et al. (2019) for which wells that do not sample the entire stratigraphy are corrected based on the oldest well of the region. We then calculated the mean crustal stretching ($\beta$ factor, Fig. 3c) for each tectonic subsidence curve based on isostatic calculation (Watts, 2001).

During the late-Permian–Early Triassic, a first phase of significant tectonic subsidence, up to 500 m, is recorded in the Maestrat Basin and on the Iberia paleomargin of the Betic basins (Salas and Casas, 1993; Van Wees et al., 1998; Salas et al., 2001; Hanne et al., 2003; Soto et al., 2019) (Fig. 3b–c). This phase is contemporaneous with the westward migration of marine deposition in the Iberian basins during the Middle Triassic (Anisian-Carnian, 240–230 Ma). Sopeña et al. (1988) CE10 argues that Tethyan rifting propagated westward inboard CE11 Iberia. The same evolution is suggested by the stratigraphy and the depositional evolution constraints from the Catalan and Basque–Cantabrian basins (Sopeña et al., 1988), and in the Aquitaine domain (Fig. 3b), although it is ill-defined for Permian times.

During the Late Triassic (220–200 Ma), the regional tectonic subsidence in all regions is found to be associated with the deposition of evaporites that spread all over Iberia, in the Betics, western Iberia, and in the Aquitaine Basin (Fig. 3a). The distribution of salt terrane in Iberia and its surroundings (Fig. 3a) highlights a very large subsiding domain for this period. A maximum mean subsidence of 700 m is inferred in the Maestrat Basin for Triassic times. The relatively rapid subsidence in the Triassic contrasts with the slower subsidence observed during the Early–Middle Jurassic. A notable exception is depicted by the slight increase in subsidence between 200 and 150 Ma in the Betics (Fig. 3b–c), consistent with rifting across the Iberia–Africa boundary (Ramos et al., 2016; Fernández, 2019).

A third Late Jurassic–Early Cretaceous phase (150–110 Ma) is marked by the increase in tectonic subsidence in the Iberian basins, coeval with the expected timing of strike-slip deformation and rifting in the Cameros (e.g., Rat et al.,

2019; Aurell et al., 2019) and Columbrets (Etheve et al., 2018) basins as well as the initiation of mantle exhumation in the Atlantic domain (Fig. 1a) (Murillas et al., 1990; Mohn et al., 2015). The most recent extension is recorded in the Aquitaine Basin at 120–100 Ma, which reflects the onset of oceanic spreading in the Bay of Biscay (Fig. 3b–c).

Subsidence analyses show thinning events in Iberia that reveal control by Tethys and Atlantic rifting (late Permian–Late Triassic) and later by the intra-Iberian–Pyrenean rift events (Late Jurassic–Early Cretaceous). In the Iberian Basin, this latter event is characterized by a relatively large and short-lived subsidence (1.5 km in 30 Myr) localized in narrow basins that suggests the strike-slip nature of the boundary between Ebro and Iberia in the Late Jurassic. The long-lasting rift evolution, however, shows an average low stretching factor of about 1.2.

## 4 Methodology

### 4.1 Previous kinematic models

We compile and implement previous kinematic models involving Iberia (Fig. 4). The objective is to establish a coherent kinematic model of Iberia that considers both the evolution of the Neotethyan and Atlantic regions. These kinematic models are either global (e.g., Müller et al., 2019), based on the assimilation of geological and geophysical information at a large scale to allow a dynamic understanding of Earth's plate tectonics but do not aim to solve regional tectonic issues such as strain partitioning between Iberia and Europe. On the other hand, regional models are focused on the reconstruction of the North Atlantic (e.g., Barnett-Moore et al., 2016; Nirrengarten et al., 2018; Peace et al., 2019b) or are interested in the reconstruction of the Alpine orogen with inferences on the kinematics of the Tethys and Adria (Schmid et al., 2008; Handy et al., 2010; Van Hinsbergen et al., 2019).

North Atlantic reconstructions use offshore geophysical constraints from the northwest Iberian margins and pay relatively little attention to the geological evolution of the Pyrenees and other orogenic domains in Iberia (e.g., Sibuet et al., 2004; Barnett-Moore et al., 2016; Nirrengarten et al., 2018). However, these models give fundamental insights into the geometry of the North Atlantic full-fit reconstruction and the timing of oceanic spreading. The nature of some magnetic anomalies in the southern North Atlantic has been the matter of considerable debate (Olivet, 1996; Sibuet et al., 2004; Vissers and Meijer, 2012; Barnett-Moore et al., 2016). Here, we adopt the reconstruction of Nirrengarten et al. (2018), who propose a model based on the re-evaluation of magnetic anomalies that are considered not oceanic before C34 (83 Ma) and therefore not suitable for kinematic studies (Nirrengarten et al., 2017).

Reconstructions of the Alpine domain (Schmid et al., 2008; Handy et al., 2010) and at a larger scale of the Tethys

https://doi.org/10.5194/se-11-1-2020

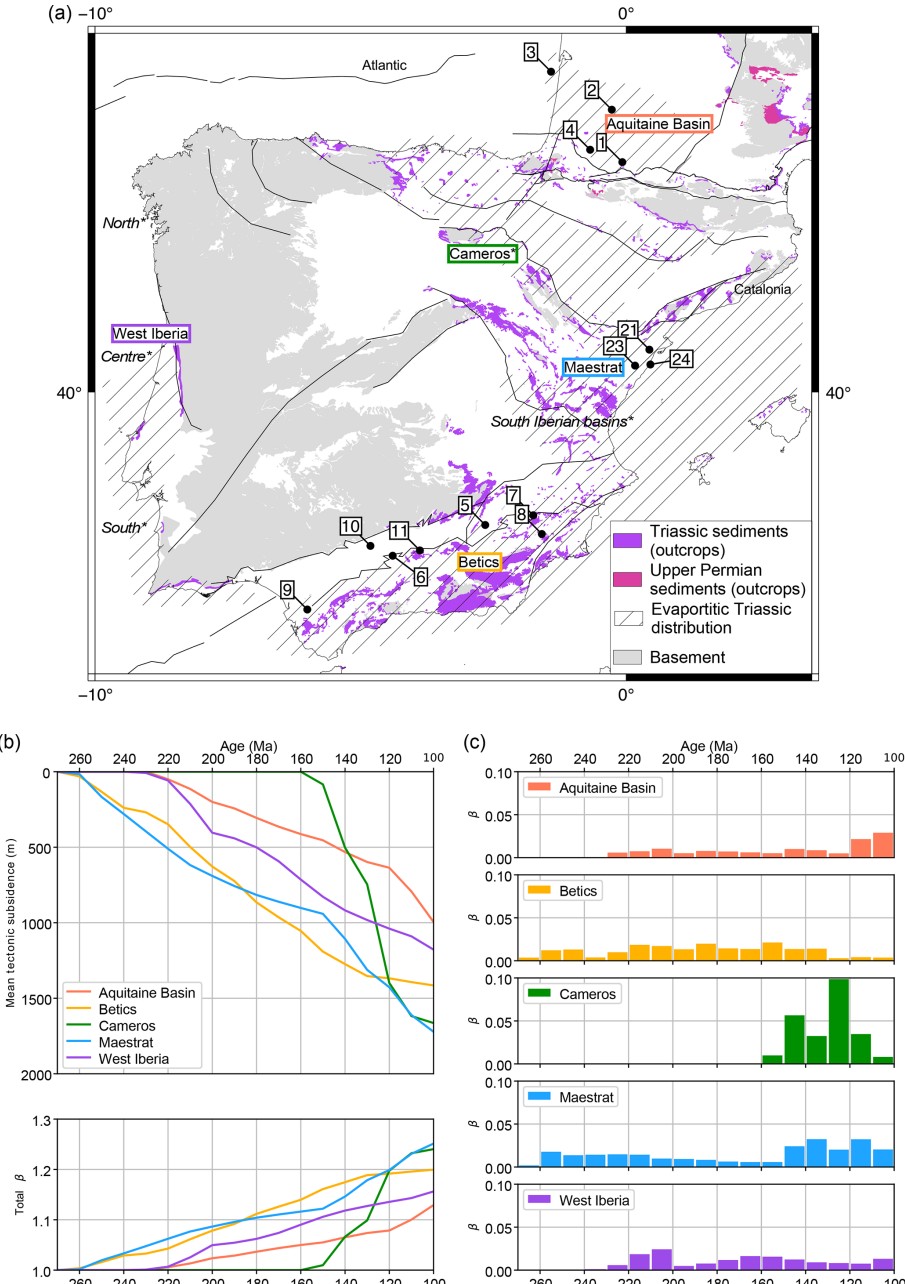

**Figure 3.** Late Permian–Triassic deposits and subsidence analyses. **(a)** Map of the upper-Permian–Triassic sediment outcrops, main depocenters, and distribution of the Upper Triassic evaporitic sequence in Iberia and southwest France. **(b)** Top: mean tectonic subsidence curves in the Aquitaine Basin (Brunet, 1984), Betics (Hanne et al., 2003), Cameros Basin (Salas and Casas, 1993; Salas et al., 2001; Omodeo-Sale et al., 2017), Maestrat Basin (Salas and Casas, 1993; Salas et al., 2001), and West Iberia (Spooner et al., 2018). See Supplement for individual tectonic subsidence curves in each region. (1) Ger1; (2) Lacquy1; (3) Sextant1; (4): Lacq301; (5) Santiago de la Espada; (6) Nueva Carteya1; (7) Rio Segura G1; (8) Espugna; (9) Betica 18-1; (10) Rio Guadalquivir; (11) Fusanta; (12) Lazaro; (13) Fuentatoba*; (14) Poveda*; (15) Cameros2*; (16) Enciso*; (17) Rollamentia*; (18) Castellijo*; (19) Molino*; (20) Yanguas*; (21) Mirambell; (22) Amposta Marino C3; (23) Salzedella; (24): Maestrazgo; (25): South Iberian Basin; (26): West Iberia south CE7; (27): West Iberia central CE8; (28): West Iberia north CE9. Synthetic wells (shown by *) are not represented on the map. Bottom: total stretching factor ($\beta$) (isostatic calculation, Watts, 2001) calculated from the mean tectonic subsidence of each region. **(c)** Incremental stretching factor ($\beta$) with a 10 Myr time step.

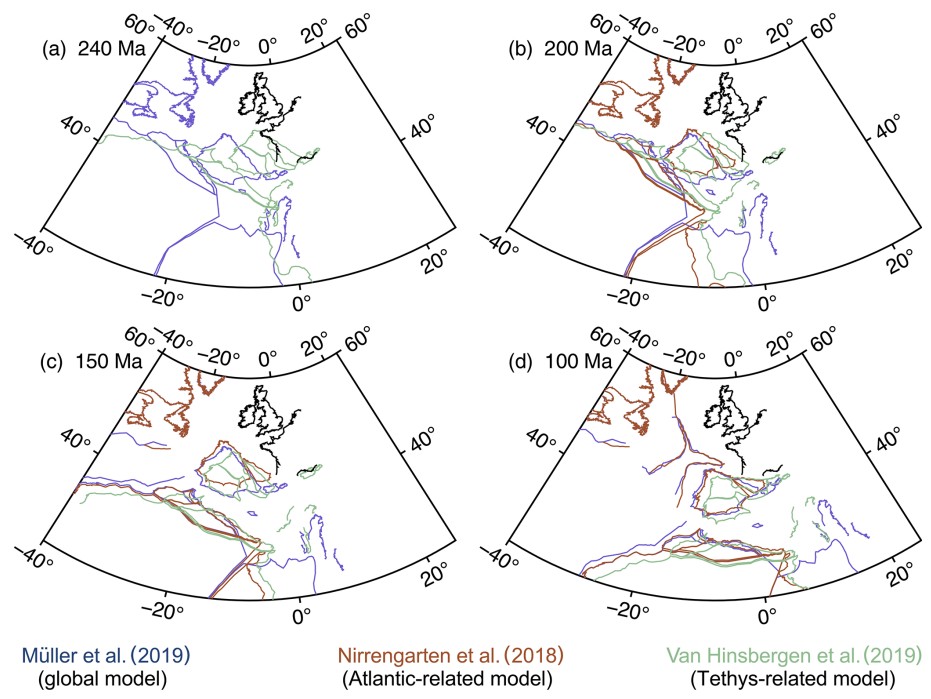

**Figure 4.** Compilation of previous global (Müller et al., 2018), Atlantic-related (Nirrengarten et al., 2018), or Tethys-related (Van Hinsbergen et al., 2019) kinematics reconstructions used in our reconstruction, at 240 Ma **(a)**, 200 Ma **(b)**, 150 Ma **(c)**, and 100 Ma **(d)**. Note that the Nirrengarten et al. (2018) reconstruction is not shown at 240 Ma as represented in their model.

domain (Van Hinsbergen et al., 2019) are keys to understanding the evolution of past oceanic domains now inverted in Alpine orogenic systems (e.g., Paleotethys, Neotethys, Meliata, Pindos, and Vardar oceans). These models however do not account for the recent reconstruction of the southern North Atlantic presented above that have an impact on the movement of Iberia of interest for our study.

## 4.2 Reconstruction workflow

A plate reconstruction from the late Permian to middle Cretaceous is presented in Figs. 5 and 6, based on kinematic modeling using GPlates version 2.1 (Müller et al., 2018). The rotation poles of the main plates are summarized in Table 1 (see also the Supplement for GPlates files).

Our compilation is as follows: (1) the reconstruction of the western Tethys [CE13] prior to the Late Jurassic is constrained by the kinematic evolution of the Mediterranean region since the Triassic from Van Hinsbergen et al. (2019), which we corrected for overlap of Iberia over western France; (2) the kinematics of Africa follows Müller et al. (2019), based on Heine et al. (2013); (3) for the Late Jurassic and Cretaceous times, we compiled rotation poles of the North America–Europe system from Barnett-Moore et al. (2016), updated from Peace et al. (2019) [TS5] for North Atlantic continental blocks (Flemish Cap, Orphan Knoll, and Porcupine Bank); (4) our reconstruction of Adria follows the model from Müller et al. (2019), which was modified to account for the

possible younger opening of the Ionian Basin (Tugend et al., 2019).

Because there is no motion during the 270–250 Ma interval for the North America and Africa plates relative to Europe (Domeier and Torsvik, 2014), we extended the full-fit of these models to 270 Ma.

These input models were then updated according to the following constraints (Table 2): (1) age of rifting, mantle exhumation, onset of oceanic spreading in the Atlantic; (2) the present-day position of ophiolite bodies and the timing of rifting, oceanic spreading and subduction for the Tethyan-related oceanic domains (Paleotethys, Neotethys, Pindos, Meliata, Vardar); (3) at 100 Ma, Iberia is close to the present-day position relative to Europe, so that the late-Mesozoic–Cenozoic Pyrenean shortening is essentially orthogonal.

## 4.3 Implementations of the pre-existing models

A critical step in determining the pre-rifting configuration is the restoration of rifted margins. Here, we adopted the reconstructed continental crust geometry of Nirrengarten et al. (2018). Polygons from the model of Nirrengarten et al. (2018) that are based on Seton et al. (2012) were re-defined such that they include new smaller polygons (continental micro-blocks) separated by deformed areas in Iberia and Adria to account for internal deformation (Fig. 1b).

The kinematics of these continental blocks (e.g., the Ebro block) has been reconstructed using geological constraints

Please note the remarks at the end of the manuscript.

**Table 1.** Rotation poles of the main plates or continental blocks used in the study. Poles with no references are from this study. See the complete list in the GPlates rotation file in Supplement. [TS4]

| Age (Ma) | Latitude | Longitude | Angle | Fixed plate [CE12] | Ref |
|---|---|---|---|---|---|
| **North America** | | | | | |
| 83.0 | 76.81 | −20.59 | 29.51 | NW AFR | Nirrengarten et al. (2018) |
| 120.4 | 66.28 | −19.82 | 54.44 | NW AFR | Nirrengarten et al. (2018) |
| 126.7 | 66.11 | −18.95 | 56.48 | NW AFR | Nirrengarten et al. (2018) |
| 131.9 | 65.95 | −18.5 | 57.45 | NW AFR | Nirrengarten et al. (2018) |
| 139.6 | 66.12 | −18.38 | 59.9 | NW AFR | Nirrengarten et al. (2018) |
| 147.7 | 66.54 | −17.98 | 62.08 | NW AFR | Nirrengarten et al. (2018) |
| 154.3 | 67.15 | −15.98 | 64.75 | NW AFR | Nirrengarten et al. (2018) |
| 170.0 | 67.09 | −13.86 | 70.55 | NW AFR | Nirrengarten et al. (2018) |
| 190.0 | 64.31 | −15.19 | 77.09 | NW AFR | Nirrengarten et al. (2018) |
| 203.0 | 64.28 | −14.74 | 78.05 | NW AFR | Nirrengarten et al. (2018) |
| 270.0 | 64.28 | −14.74 | 78.05 | NW AFR | Nirrengarten et al. (2018) |
| **Flemish Cap** | | | | | |
| 112.0 | 90.0 | 0.0 | 0.0 | NAM | Nirrengarten et al. (2018) |
| 140.0 | 45.28 | −53.47 | 20.03 | NAM | Nirrengarten et al. (2018) |
| 160.0 | 44.65 | −54.79 | 18.83 | NAM | Nirrengarten et al. (2018) |
| 200.0 | 42.8694 | −54.8782 | 17.7035 | NAM | Nirrengarten et al. (2018) |
| 270.0 | 42.8694 | −54.8782 | 17.7035 | NAM | Nirrengarten et al. (2018) |
| **Europe** | | | | | |
| 79.1 | 63.4 | 147.75 | −18.48 | NAM | Nirrengarten et al. (2018) |
| 120.0 | 68.01 | 153.59 | −21.01 | NAM | Nirrengarten et al. (2018) |
| 200.0 | 71.41 | 152.6 | −23.68 | NAM | Nirrengarten et al. (2018) |
| **West Iberia (Galicia)** | | | | | |
| 86.0 | −40.18 | 170.57 | 10.07 | EUR | |
| 120.0 | −47.9216 | 179.85 | 22.8717 | EUR | |
| 135.0 | −49.5226 | −177.8405 | 27.0209 | EUR | |
| 140.0 | −49.695 | −177.3636 | 27.3079 | EUR | |
| 150.0 | −50.3915 | −177.3963 | 27.8356 | EUR | |
| 270.0 | 50.9548 | 5.1656 | −28.4002 | EUR | |

| Age (Ma) | Latitude | Longitude | Angle | Fixed plate | Ref |
|---|---|---|---|---|---|
| **Ebro block** | | | | | |
| 0.0 | 90.0 | 0.0 | 0.0 | EUR | |
| 20.0 | 90.0 | 0.0 | 0.0 | EUR | |
| 66.0 | 40.1807 | 8.7194 | 5.606 | EUR | |
| 84.0 | −41.0445 | 136.3388 | 2.8866 | EUR | |
| 94.0 | −41.0445 | 136.3388 | 2.8866 | EUR | |
| 118.0 | −45.8785 | 165.4893 | 4.2288 | EUR | |
| 145.0 | −62.2857 | 170.8607 | 4.5896 | EUR | |
| 270.0 | −58.2298 | −176.5352 | 11.1943 | EUR | |
| **Southwest Iberia** | | | | | |
| 84.0 | −28.0047 | −158.3184 | −0.9344 | WIB | |
| 100.0 | −36.5872 | 156.7264 | 0.6206 | WIB | |
| 120.0 | 29.0472 | 91.4916 | −0.257 | WIB | |
| 130.0 | −43.3609 | 171.3962 | 10.9309 | WIB | |
| 145.0 | −45.4085 | 167.4179 | 8.8479 | WIB | |
| 150.0 | 47.1092 | −9.648 | −7.8087 | WIB | |
| 270.0 | 45.8739 | −8.1058 | −9.9091 | WIB | |
| **Adria** | | | | | |
| 100.0 | −26.5 | −166.58 | 7.73 | APU | Müller et al. (2019) |
| 120.0 | −26.5 | −166.58 | 7.73 | APU | Müller et al. (2019) |
| 250.0 | −26.5 | −166.58 | 7.73 | APU | Müller et al. (2019) |
| 270.0 | −26.5 | −166.58 | 7.73 | APU | |
| **Northwest Africa** | | | | | |
| 0.0 | 90.0 | 0.0 | 0.0 | NE AFR | |
| 110.0 | 90.0 | 0.0 | 0.0 | NE AFR | Heine et al. (2013) |
| 145.0 | 25.21 | 5.47 | 2.87 | NE AFR | Heine et al. (2013) |
| 231.0 | 25.21 | 5.47 | 2.87 | NE AFR | Heine et al. (2013) |
| 270.0 | 25.21 | 5.47 | 2.87 | NE AFR | Heine et al. (2013) |

**Table 2.** Geodynamic and timing constraints used in the kinematic reconstruction model.

| Domain | Area | Event/kinematics | Age (Ma) | References |
|---|---|---|---|---|
| Central Europe | Germanic–Polish Basin | Continental rifting | 270 (?) to 250 | Evan et al. (1990); Van Wees et al. (2000); Evans et al. (2003); Jackson et al. (2019) |
| | Tornquist Zone | Right-lateral<br>Left-lateral<br>Right-lateral | Carboniferous to 250 (?)<br>170<br>145 to 125 | Phillips et al. (2019)<br>Phillips et al. (2019)<br>Hippolyte (2002); Phillips et al. (2019) |
| Western Europe (France) | Aquitaine Basin | Continental rifting | 270 (?) to 145; 125 to 94 | Curnelle (1983) TS6; Brunet (1984) TS7; Biteau et al. (2006) TS8; Serrano et al. (2006) TS9 |
| | Pyrenees | Continental rifting - left lateral<br>Hyper-extended rifting – left-lateral | 270 (?) to 145<br>125 to 94 | Curnelle (1983); Lucas (1995) TS10<br>Vielzeuf and Kornprobst (1984) TS11; Golberg and Leyreloup (1990) TS12; Lagabrielle et al. (2010) |
| Iberia | Basque–Cantabrian Basin | Left-lateral | 140 to 120 | Quintana et al. (2015) TS13; Zamora et al. (2017) TS14; Nirrengarten et al. (2018) |
| | Ebro Basin | Continental rifting | 270 (?) to 145 | Vargas et al. (2009) TS15 |
| | Iberian Range | Continental rifting – left lateral<br>Hyper-extended rifting | 270 (?) to 145<br>150 to 120 | Salas and Casas (1993)<br>Salas and Casas (1993); Arche and Gomez (1996) TS16; Salas et al. (2001); Omodeo-Sale et al. (2017); Rat et al. (2019) |
| Southern North Atlantic | W Galicia | Continental rifting | 200 to 145 | Murillas et al. (1990) |
| | | Lithospheric mantle exhumation<br>Initiation of oceanic spreading | 135–115<br>133–100 | Mohn et al. (2015)<br>Olivet (1996); Strivastava et al. (2000) TS17; Schettino and Turco (2009) TS18); Whitmarsh and Manatschal (2012) TS19 |
| | Bay of Biscay | Mantle exhumation<br>Oceanic spreading | 160 to 130<br>124–112 to 83 | Thinon et al. (2001) TS20; Tugend et al. (2014)<br>Thinon (2002) TS21; Sibuet et al. (2004); Tugend et al. (2015) |
| | Southwest Iberia | Continental rifting<br>Mantle exhumation<br>Initiation of oceanic spreading | 200 to 161<br>145<br>135–133 (Gorringe Bank) | Murillas et al. (1990)<br>Sallàres et al. (2013)<br>Sallàres et al. (2013) |
| North Sea | Arctic rift system | Continental rifting | 290 (?) to 200 | Evans et al. (2003) |
| | Rockall–Porcupine | Continental rifting | 230 (?) to 112 | Evans et al. (2003) |
| | Orphan | Continental rifting | 270 to 112 | Nirrengarten et al. (2018); Hassan et al. (2019) TS22; Sandoval et al. (2019) |
| Tethys and peri-Tethys | S Alpine Tethys | Continental rifting<br><br>Breakup | 270 to 220<br><br>180 | Stampfli and Borel (2002); Schmid et al. (2008); Sciscani and Esestime (2017)<br>Schmid et al. (2008); Puga et al. (2011); Marroni et al. (2017) |
| | N Alpine Tethys | Oceanic spreading | 170–161 | Bill et al. (2001); Schaltegger et al. (2002) |
| | Paleotethys | Subduction | Early Carboniferous to 200 | Stampfli et al. (2001); Stampfli and Borel (2002); Evans et al. (2003) |
| | Neotethys sensu stricto | Oceanic spreading<br>Subduction | Early Permian (?)<br>from 156 | Van Hinsbergen et al. (2019)<br>Schmid et al. (2008); Van Hinsbergen et al. (2019) |
| | Ionian | Continental rifting<br>Oceanic spreading | 270 to 200 (?)<br>onset at 180 ? | Muttoni et al. (2001); Tugend et al. (2019)<br>Tugend et al. (2019) |
| | Vardar | Oceanic spreading<br>Subduction | 180 to 160 (?)<br>145 to 110 | Channell and Kozur (1997)<br>Channell and Kozur (1997) |
| | Pindos | Oceanic spreading<br><br>Subduction | 250 to 200<br><br>From Late Cretaceous | Channel and Kozur (1997); Stampfli et al. (2001); Schmid et al. (2008)<br>Channell and Kozur (1997) |
| | Meliata | Oceanic spreading<br><br>Subduction | 220 to 200<br><br>180 to 160 (?) | Channel and Kozur (1997); Stampfli et al. (2001); Schmid et al. (2008)<br>Channell and Kozur (1997) |
| Central Atlantic | | Continental rifting<br>Oceanic spreading | 250 to 200<br>190 to 175 | Kneller et al. (2012) TS23<br>Labails et al. (2010); Olyphant et al. (2017) |

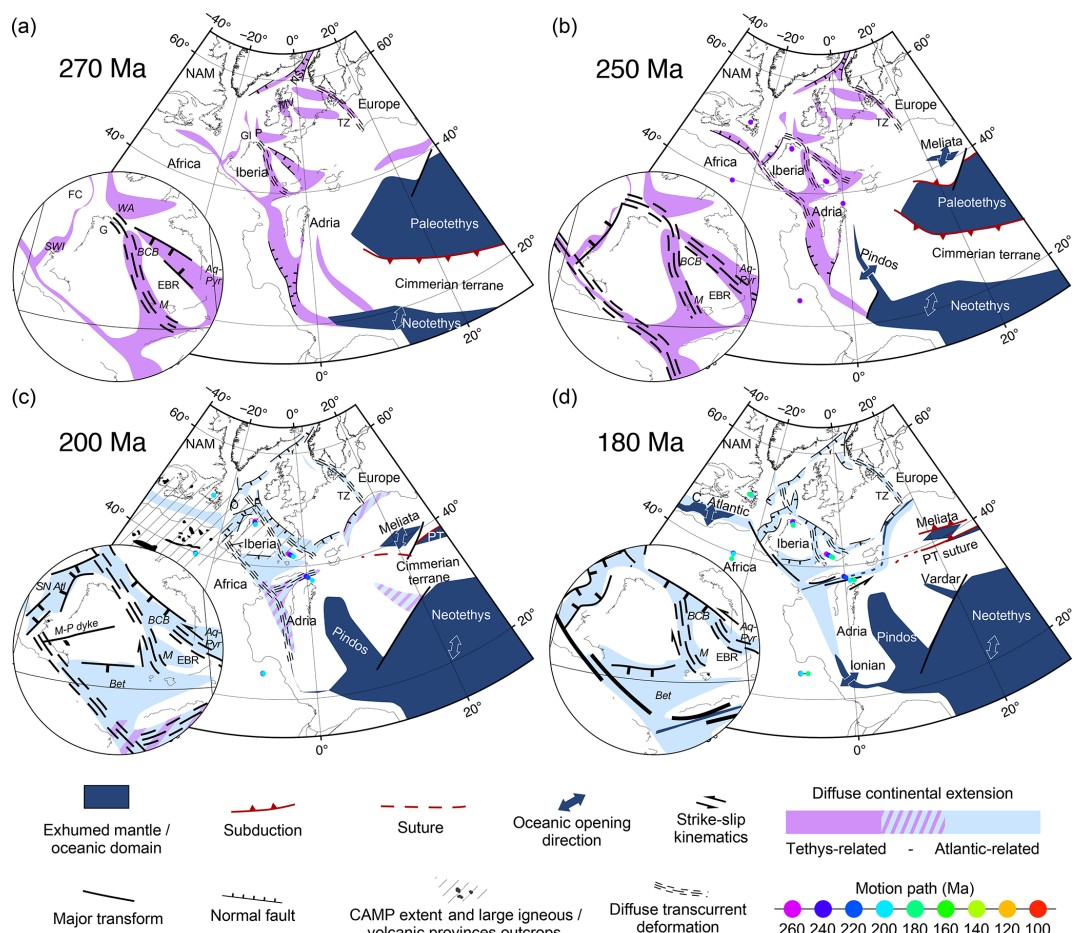

**Figure 5.** Large-scale reconstruction of the Tethys–Atlantic area for the 270 Ma (**a**), 250 Ma (**b**), 200 Ma (**c**), and 180 Ma (**d**) periods. Maps are in orthographic projection. Closeup shows the evolution of the Iberia plate and the Ebro continental block. Colored background shows diffuse or ill-defined deformation (extension or transtension) in the continents, associated with the evolution of the Tethys (Paleo- and Neotethys, purple) or the Atlantic (blue). At 200 Ma, the hatched area represents the CAMP extent. Aq-Pyr: Aquitaine–Pyrenean Basin; BCB: Basque–Cantabrian Basin; Bet: Betics Basin; EBR: Ebro; FC: Flemish Cap; G: Galicia; GI: Galicia Interior Basin; M: Maestrat; M-P: Messejana–Plascencia; MV: Midland Valley rift; NS: North Sea rift; O: Orphan Basin; P: Porcupine Basin; PT: Paleotethys; SN Atl: southern North Atlantic; SWI: southwest Iberia; TZ: Tornquist Zone; WA: Western Approaches.

inferred from the tectono-sedimentary evolution of intra- and peri-Iberian basins (see Sect. 3 and Fig. 3), which allows defining periods of deformation and subsidence related to extension or transcurrent deformation.

Because a full-fit reconstruction in southwest Iberia leads to significant overlapping between the Flemish Cap and Galicia, we use the Nazaré Fault (Pereira et al., 2017) to segment western Iberia. This allows us to minimize the overlap of northwest Iberia (Galicia) over the Flemish Cap or to have a gap between southwest Iberia and Newfoundland.

# 5   Kinematics of Iberia between the Atlantic and Tethys

## 5.1   Permian–Late Triassic (270–200 Ma)

The Neotethys Ocean opening initiated in the early Permian in the northern Gondwana margin, resulting in the northward drift of the Cimmerian terrane and the subduction of the Paleozoic Paleotethys Ocean (Stampfli et al., 2001; Stampfli and Borel, 2002). This occurred contemporaneously with the establishment of the Carboniferous-Permian magmatic activity in the North Sea rift and Midland Valley rift areas (Evans et al., 2003; Heeremans et al., 2004; Upton et al., 2004).

As the Neotethys rift propagated westwards, diffuse continental rifting took place in the whole of western Europe defined by the position of the Paleozoic Variscan and Caledonian orogenic belts in the west, the Tornquist suture in

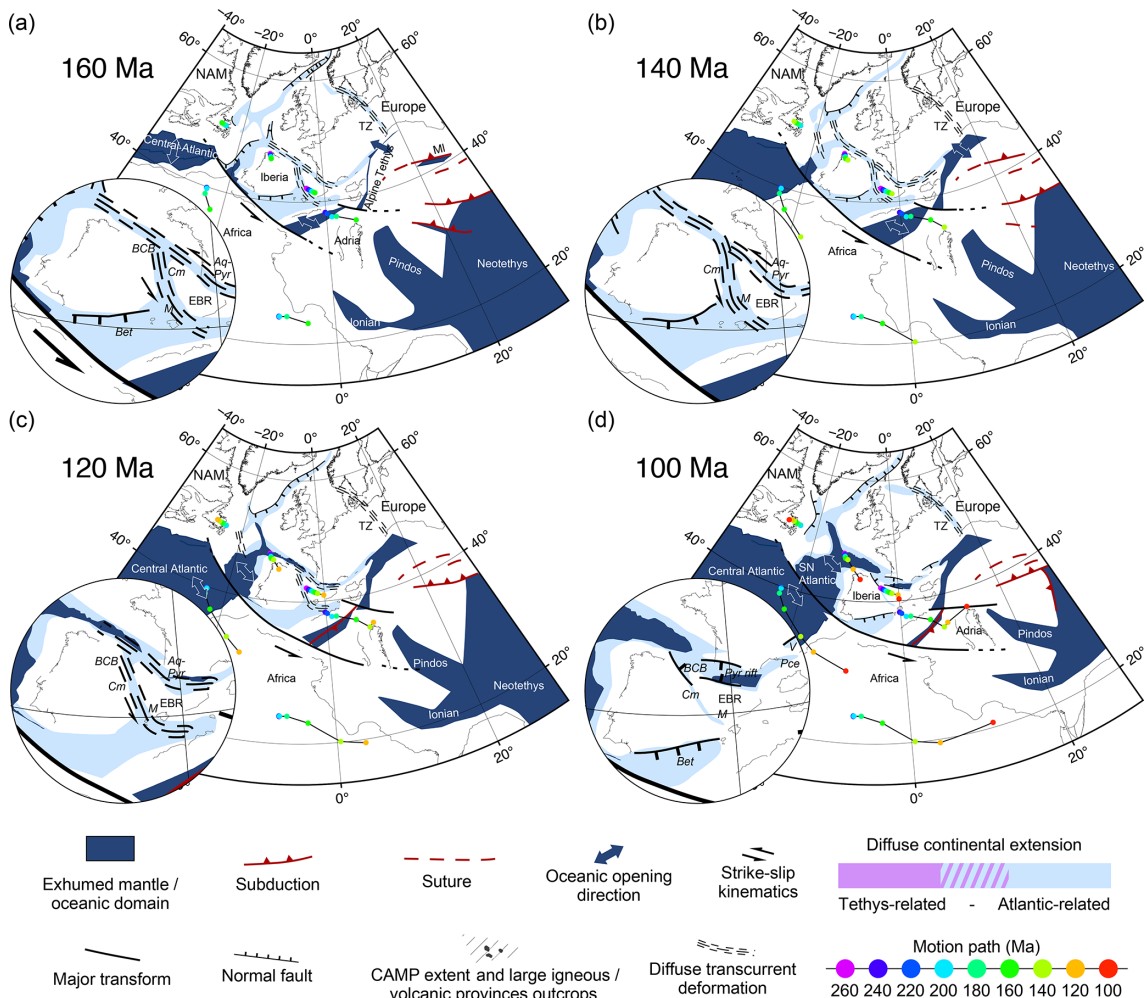

**Figure 6.** Large-scale reconstruction of the Tethys–Atlantic area for the 160 Ma (**a**), 140 Ma (**b**), 120 Ma (**c**), and 100 Ma (**d**) periods. Maps are in orthographic projection. Closeup shows the evolution of the Iberia plate and the Ebro continental block. Colored background shows diffuse or ill-defined deformation (extension or transtension) in the continents, associated with the evolution of the Tethys (Paleo- and Neotethys, purple) or the Atlantic (blue). Aq-Pyr: Aquitaine–Pyrenean Basin; BCB: Basque–Cantabrian Basin; Bet: Betics Basin; Cm: Cameros; EBR: Ebro; M: Maestrat; Ml: Meliata; Pce: Provence Basin; V: Valaisan Basin; TZ: Tornquist Zone.

the east and a diffuse transtensional transfer zone along the Africa–Iberia–Adria boundary (Fig. 5a). This is recorded by several late-Permian rift domains located in the southern North Atlantic (Rasmussen et al., 1998; Leleu et al., 2016), in the Adriatic (Scisciani and Esestime, 2017) in the North Sea (Hassaan et al., 2020), in the Germanic rift basins, including the Zechstein Basin (Evans, 1990; Van Wees et al., 2000; Jackson et al., 2019), and in Iberia (Figs. 2, 3, and 5a). A recent study (Sandoval et al., 2019) also showed a high pre-Early Jurassic thinning in North Atlantic basins (e.g., Galicia Interior, Porcupine, and Orphan basins).

Back-arc extension associated with the subduction of the Paleotethys (Van Hinsbergen et al., 2019) (Fig. 5b) triggered the extension and formation of oceanic basins in the Pindos and Meliata domains during the Early (250 Ma) and Late Triassic (Carnian, 220 Ma), respectively (Channell and Kozur,

1997; Stampfli et al., 2001). As proposed by Schmid et al. (2008), the Pindos Ocean was probably a western branch of the Neotethys rather than a unique ocean. The strike-slip reactivation of the Tornquist Zone could also be a far-field effect of Paleotethys closure, as suggested by Phillips et al. (2018).

During the Late Triassic–Early Jurassic (Fig. 5c–d) the opening of the Ionian Basin (Tugend et al., 2019) triggers northward displacement of Adria relative to Iberia and Africa and induced transtension between Adria and Iberia. This is consistent with Triassic basins of eastern Betics and Catalonia that developed at the emplacement of the future Alpine Tethys, the rifting of which started from the Late Triassic (220 Ma) (Stampfli and Borel, 2002; Schmid et al., 2008). The large rift-related subsidence in the Iberian basins (Fig. 3b) is kinematically consistent with the stretching lin-

eations documented from Triassic strata (Soto et al., 2019). Ebro is already individualized from Iberia and moved eastwards relative to Iberia and Europe through right-lateral and left-lateral strike-slip movements, respectively.

## 5.2 Early Jurassic (200–160 Ma)

This period marks a gradual change from Tethyan-dominated to Atlantic-dominated tectonism in Iberia. As the Neotethys propagated in the Vardar Ocean, the Pindos and Meliata oceans started to close (Fig. 5c) (Channell and Kozur, 1997). Major dynamic changes occurred with the CAMP event (Olsen, 1997; Marzoli et al., 1999; McHone, 2000; Leleu et al., 2016; Peace et al., 2019b) that led to breakup in the central Atlantic Ocean during the 190–175 Ma interval (Pliensbachian-Toarcian) (Fig. 5c–d) according to Labails et al. (2010) and Olyphant et al. (2017). The propagation of the central Atlantic rift northwards caused extension to propagate in the southern North Atlantic (Murillas et al., 1990; Leleu et al., 2016) and laterally, eastward in the Alpine Tethys (Schmid et al., 2008; Marroni et al., 2017) by some reactivation of Triassic Neotethyan rift structures. Evidence for nearly synchronous intrusions of MORB CE14-type gabbro, in a western branch of the Alpine Tethys, is described at 180 Ma in the internal zones of eastern Betics (Puga et al., 2011), associated with the rapid subsidence in the Betics (Fig. 3b). However, whether this is related to incipient oceanic spreading or magmatism in a hyper-extended margin is controversial. By contrast, both the thermal and stratigraphic evolutions (also Fig. 2) suggest that central Iberia remained little affected by the propagation of the Early Jurassic Atlantic rift Iberian basins (Aurell et al., 2019; Rat et al., 2019). A kinematic change from oblique to orthogonal E–W extension in the Alpine Tethys is marked by the onset of oceanic spreading between the Bajocian–Bathonian (170–166 Ma) and the Oxfordian (161 Ma) as suggested by the ages of MORB magmatism in the Alps (Schaltegger et al., 2002) and first post-rift sediments (Bill et al., 2001). As such the Jurassic Alpine Tethys has temporal and genetic affinities with the Atlantic Ocean evolution rather than the Neotethys. The required differential movement between the opening of the Alpine oceanic domains, and the central Atlantic and the closure of the Neotethys and short-lived Vardar Oceans from 160 Ma onward induced the reactivation of the former diffuse transfer zone between Iberia and Africa into a localized transform plate boundary (Fig. 6a).

## 5.3 Late Jurassic–Early Cretaceous (160–100 Ma)

A major tectonic change occurred in the Late Jurassic–Early Cretaceous when the southernmost North Atlantic successfully rifted the continental domain located offshore of southwest Iberia in present-day coordinates (between 160 and 100 Ma; Fig. 6), as recorded by mantle exhumation and subsequent oceanic spreading at 147 and 135–133 Ma, respec-

tively, in Gorringe Bank (Sallarès et al., 2013). Oceanic opening then migrated northward, attested by mantle exhumation and oceanic offshore of southwest Galicia between 139.8 and 129.4 Ma (Mohn et al., 2015) and 121–112 Ma (Bronner et al., 2011; Vissers and Meijer, 2012) (Fig. 6b). At that time, the east-directed movement of Iberia relative to Ebro induced left-lateral transtensional faulting in a corridor shaped by the Iberian basins (Tugend et al., 2015; Aurell et al., 2019; Rat et al., 2019). We further infer a residual strike-slip movement between Ebro and Europe in the Pyrenean basins until the mid-Cretaceous (118 Ma) when the Bay of Biscay opened and the rotation of Iberia occurred (Sibuet et al., 2004; Barnett-Moore et al., 2016). The eastwards motion of Iberia relative to Adria resulted in the closure of the southern Alpine Tethys (Fig. 6c). Eastward rotation of Africa induces subduction along the northern Neotethyan margin (Schmid et al., 2008) (Fig. 6b–d).

Until 120 Ma (Early Cretaceous) eastward accommodation space is constantly created by the formation of rift segments in the southwest Alpine domain (Valaisan domain and southeast basins of France) and then the Provence domains (Tavani et al., 2018). In the southern part of the Western Alps, reactivation of Tethyan normal faults are shown to be Late Jurassic–Early Cretaceous in age (Tavani et al., 2018). At 110 Ma, deformation migrates in the South Provence Basin making a straighter continuity of the Pyrenean system toward the east (Tavani et al., 2018).

## 6 Implications for strike-slip movements and the Europe–Iberia plate boundary

### 6.1 Amount of strike-slip displacement

Table 3 summarizes the timing, amounts, and sense of strike-slip component of the Ebro kinematics relative to Europe and Iberia inferred from our model. Our reconstructions suggest a total left-lateral strike-slip movement of 278 km between Europe and Ebro. In total, 90 km were accommodated during the late-Permian–Triassic period (Fig. 5a-c, 270–200 Ma), and CE15 86 km were accommodated during the Jurassic (Figs. 5c–d and 6a–b, 200–140 Ma). We quantify 99 and 19 km for the 140–120 and 120–100 Ma time intervals, respectively, leading to a total of 128 km of strike-slip movement during the Lower Cretaceous, in the range of amounts deduced from offshore and onshore geological observations (Olivet, 1996; Canérot, 2016). By 118 Ma, most of the strike-slip faulting is terminated as extension became orthogonal and Ebro is close to its present-day position (Jammes et al., 2009; Mouthereau et al., 2014). The maximum strain rate of $5 \, \text{km} \, \text{Myr}^{-1}$ is obtained for the 140–120 Ma time interval, revealing progressive strain localization in the Pyrenean basins before mantle exhumation (Jammes et al., 2009; Lagabrielle et al., 2010; Mouthereau et al., 2014; Tugend et al., 2014).

**Table 3.** Quantification of strike-slip displacement between the European and Ebro and between the Iberia (Galicia) and Ebro.

| Age (Ma) | Iberia–Ebro | | | | Europe–Ebro | | |
|---|---|---|---|---|---|---|---|
| | Amount (km) | Rate (km Ma$^{-1}$) | Direction | | Amount (km) | Rate (km Ma$^{-1}$) | Direction |
| 270–250 | 12 | 0.6 | right-lateral | | 16 | 0.8 | left-lateral |
| 250–200 | 33 | 0.7 | right-lateral | | 74 | 1.5 | left-lateral |
| 200–180 | 17 | 0.9 | right-lateral | | 19 | 1.0 | left-lateral |
| 180–160 | 5 | 0.3 | right-lateral | | 24 | 1.2 | left-lateral |
| 160–140 | 4 | 0.2 | left-lateral | | 43 | 2.2 | left-lateral |
| 140–120 | 62 | 3.1 | left-lateral | | 99 | 5.0 | left-lateral |
| 120–100 | 179 | 9.0 | left-lateral | | 19 | 1.0 | left-lateral |
| Total | 67 km (right-lateral) | | | | | | |
| | 245 km (left-lateral) | | | | 278 km (left-lateral) | | |

The Iberia–Ebro boundary has played as right-lateral and left-lateral kinematics. The rapid eastward displacement of Ebro during the late-Permian to Late Jurassic period (Figs. 5 and 6) induces a total of 67 km (12, 33, 17, and 5 km during the 270–250, 250–200, 200–180, and 180–160 Ma time interval, respectively) right-lateral strike-slip between Ebro and Iberia (i.e., Galicia). This displacement has been partitioned with extension within the Iberian basins along a NW-directed intra-continental deformation corridor. This is consistent with stretching markers in Triassic rocks in this area (Soto et al., 2019). From 160 to 100 Ma, the northward propagation of the central Atlantic spreading ridge into the southern North Atlantic resulted in a net left-lateral slip of 245 km and increasing strain rates of up to 9 km Myr$^{-1}$, indicating that the southern Ebro boundary became the main tectonic boundary in Iberia, accommodating eastwards displacement of Iberia into the Alpine Tethys region.

The cumulated left-lateral displacement from both rift systems, corrected for the right-lateral displacement in the Iberian basins, is 456 km, consistent with the absolute 400–500 km required from the closure of the Atlantic between Iberia and Newfoundland.

## 6.2 Strike-slip structures in the intra-Iberian basins

Despite the requirement of 245 km left-lateral strike-slip displacement along the Iberia–Ebro boundary from 160 to 100 Ma, there is no simple geological evidence in support of a unique major crustal-scale fault in the Iberian Range–Basque–Cantabrian Basin system.

Several studies have suggested that a left-lateral shear zone can be recognized along the Iberian Range and the Basque–Cantabrian rifts system. Geological evidence includes the High Tagus Fault in the Iberian Range (Aldega et al., 2019; Aurell et al., 2019) and the Ventaniella Fault in the Basque–Cantabrian region (e.g., Tavani et al., 2011). The latter fault is often considered in recent reconstructions to accommodate the Iberia–Ebro movement alone (Tugend et al., 2015;

Nirrengarten et al., 2018). However, the estimated left-lateral displacement along the Ventaniella Fault is only in the order of magnitude of a few kilometers (Tavani et al., 2011) and therefore cannot be used as a North Pyrenean Fault equivalent.

In the Basque–Cantabrian Basin, the Ventaniella Fault is part of a NW–SE fault system that acted as left-lateral shear zone during the Late Jurassic–Early Cretaceous and has been subsequently inverted with a right-lateral kinematic during the Cenozoic (De Vicente et al., 2011; Tavani et al., 2011; Cámara and Flinch, 2017). These faults have a Triassic origin (Tavani and Granado, 2015). Tectonic activity along these faults gets younger NE-ward (Ubierna fault: Late Jurassic-Early Cretaceous; Zamanza–Oña fault: Early–Middle Cretaceous; salt tectonics in the center of the Basin, Cámara and Flinch, 2017). We suggest the Iberia–Ebro displacement to have possibly been distributed along these structures.

The role of the weak Triassic evaporites in efficiently decoupling deformation in the pre-salt basement from the thin-skinned extension in sedimentary cover has been emphasized largely in the Pyrenees (e.g., Grool et al., 2019; Duretz et al., 2019; Jourdon et al., 2020; Lagabrielle et al., 2020). Salt tectonics has also been suggested to have been particularly significant from the Jurassic through the Early Cretaceous in Mesozoic basins that shaped the NW-directed boundary between Ebro and Iberia, including the Basque–Cantabrian Basin (Cámara and Flinch, 2017), Parentis Basin (Ferrer et al., 2012), Cameros Basin (Rat et al., 2019), and Maestrat Basin (Vergés et al., 2020). The surface expression of the crustal strike-slip movements is inferred to have been limited in supra-salt layers.

The mechanism responsible for the independent movement of Ebro relative to Europe and Iberia prior to the opening of the southern North Atlantic remains unclear. The most likely hypothesis is that before the opening of the Alpine Tethys, the Ebro continental block was related to the regional eastward rotation of the Africa–Adria system. This rotation caused Ebro to move eastward, accommodating left-lateral

and right-lateral strike-slip kinematics in the Pyrenean and Iberian basins, respectively.

## 7 Conclusions

To revolve CE16 several long-lasting problems of the Mesozoic kinematics of Iberia, we propose to better account for deformation associated with late-Permian–Triassic rifting and the role of the Ebro continental block in accommodating complex strain partitioning along the Iberia–Europe plate boundary and replace Iberia in a refined plate reconstruction between the Atlantic and Tethys domains. We show that (1) left-lateral strike-slip movement did occur in the Pyrenees from the late Permian to the Early Cretaceous but ended as the Bay of Biscay opened; (2) late-Permian–Triassic extension in the Atlantic and Iberia (including Ebro) is key to quantifying the strike-slip movement in Iberia that is otherwise not well resolved from the geological constraints in Iberian basins and from full-fit reconstructions in the Jurassic. Salt tectonics that decouples syn-rift Iberian basins' evolution from their basement likely explains the lack of geological constraints.

The diffuse deformation across the Iberia–Europe plate boundary prior to oceanic spreading in the Bay of Biscay and hyper-extension in the Pyrenees appears to result mainly from transcurrent deformation partitioned with subordinate fault-perpendicular extension. The major intra-Iberia NW-trending strike-slip fault system outlined by spatially disconnected rift basins (Basque–Cantabrian, Cameros, Maestrat, and Columbrets basins) played a significant role in the Late Jurassic–Early Cretaceous, in addition to the North Pyrenean rift system.

By integrating the position of Iberia in the Tethyan and Atlantic evolution and propagating the effect of the eastward movement of Iberia into the Alpine Tethys, our reconstructions further imply that (1) Ebro was part of Adria before the onset of the Alpine Tethys opening, (2) the southern Alpine Tethys closed in the Early Cretaceous (145 to 100 Ma), and (3) the boundary between Iberia and Africa localized as a transform plate boundary at 160 Ma, connecting the Alpine oceanic domains with the central Atlantic.

*Data availability.* . TS24

*Supplement.* The supplement related to this article is available online at: https://doi.org/10.5194/se-11-1-2020-supplement.

*Author contributions.* This article was mostly written by PA and FM. PA carried out the compilation of data, interpretation, and kinematic model and figures, in tight collaboration with FM. The text benefits from the expertise and contribution of EM and RA.

*Competing interests.* The authors declare that they have no conflict of interest.

*Acknowledgements.* This study is part of the OROGEN scientific project (Total/CNRS-INSU/BRGM) as a post-doctoral grant of Paul Angrand. We acknowledge the members of the OROGEN scientific project for support and discussions, in particular S. Calassou TS25, O. Vidal, I. Thinon, L. Moen-Maurel, M. Ford, L. Jolivet, G. Manatschal and G. Frasca. We thank Alexander L. Peace and an anonymous referee for very constructive comments and suggestions that led us to improve the paper.

*Financial support.* This research has been supported by the NAME OF FUNDER (grant no. GRANT AGREEMENT NO). TS26

*Review statement.* This paper was edited by Mark Allen and reviewed by Alexander L. Peace and one anonymous referee.

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

## Remarks from the language copy-editor

CE1    Is this one affiliation or two (Toulouse and Paul Sabatier)?

CE2    Spelling adjusted. Please confirm.

CE3    Please note that this paper has been copy-edited according to American English standards.

CE4    Should this be "the Ebro block" or "the Ebro Basin" throughout?

CE5    Do you mean the northern part of the Wester Approaches or is "North Western Approaches" a set term/proper noun?

CE6    Please check the text here. This paragraph is almost identical to what precedes it. Should it be deleted or are other adjustments necessary?

CE7    Please confirm.

CE8    Please confirm.

CE9    Please confirm.

CE10    Please confirm this change or clarify the structure of the original sentence.

CE11    This is a little unclear. Do you mean into Iberia?

CE12    Should the abbreviations in this and the other "Fixed plate" column be defined or are they well known?

CE13    "Western" is capitalised in the title and my understanding is that "Western Tethys" is a proper noun. Please let us know whether this is the case or whether it should be "western Tethys" throughout. Thank you!

CE14    Should this abbreviation be defined for clarity or is it well known?

CE15    The changes here and at the beginning of the sentence were made to avoid beginning a sentence with a numeral. Please let us know if you would prefer different wording

CE16    Should this be "resolved"?

## Remarks from the typesetter

TS1    Please add city.

TS2    The composition of Figs. 1 and 4 has been adjusted to our standards. This also includes language adjustments to Fig. 1 and 4–6.

TS3    Not listed in the references.

TS4    The references were moved into the table for consistency with the other table.

TS5    a or b?

TS6    Not listed in the references.

TS7    Used several times but not listed in the references.

TS8    Used several times but not listed in the references.

TS9    Used several times but not listed in the references.

TS10    Not listed in the references.

TS11    Not listed in the references.

TS12    Not listed in the references.

TS13    Not listed in the references.

TS14    Not listed in the references.

TS15    Not listed in the references.

TS16    Not listed in the references.

TS17    Not listed in the references.

TS18    2011?

TS19    Not listed in the references.

TS20    Not listed in the references.

TS21    Not listed in the references.

TS22    Not listed in the references.

TS23    Not listed in the references.

TS24    Please provide a statement on how your underlying research data can be accessed. If the data are not publicly accessible, a detailed explanation of why this is the case is required. The best way to provide access to data is by depositing them (as well as related metadata) in reliable public data repositories, assigning digital object identifiers (DOIs), and properly citing data sets as individual contributions. Please indicate if different data sets are deposited in different repositories or if data from a third party were used. Additionally, please provide a reference list entry including creators, title, and date of last access. If no DOI

is available, assets can be linked through persistent URLs to the data set itself (not to the repositories' home page). This is not seen as best practice and the persistence of the URL must be secured.

TS25    Please provide full first names.

TS26    Please note that there is funding information given in the acknowledgements, but you did not indicate any funding upon manuscript registration. Therefore, we were not able to complete the financial support statement. Please provide the missing information and double-check your acknowledgements to see whether repeated information can be removed from the acknowledgements. Thanks.

TS27    Please add volume.

TS28    Please add volume.

TS29    Please add more information.

TS30    Please check author list. It should be displayed as follows: author comma initial(s).

TS31    Please add last access date and check link.

TS32    Please add volume.

TS33    Please add more information.

TS34    Please confirm addition.

TS35    Please add more information.

TS36    Please add volume.

TS37    Please add more information.

TS38    Please add volume.

TS39    Please add more information.

TS40    Please add page range or DOI.

TS41    Please add more information.

TS42    Please add all authors.