# Peer review of "A reconstruction of Iberia accounting for W-Tethys/N-Atlantic kinematics since the late Permian-Triassic"

_Solid Earth, 2020_

## Referee Comment (RC1) · Anonymous Referee #1 · 18 Mar 2020

This is a short paper dealing with the long lasting problem of the Mesozoic kinematics of Iberia. Here the authors revise the the Permo-Triassic rifting stage in Iberia and surrounding regions, and propose that including this stage into the puzzle may help in reconciling geological evidence and plate kinematic models. In detail, the authors suggest that Iberia cannot be considered an integer plate but, rather, it must be separated into the Ebro and Western Iberia blocks, which is in agreement with most of the recently published works on this topic/area.

The work is well written and well illustrated.

There are some minor points that should be addressed and a major issue.

[Figure]

Major point Hundreds of km of Mesozoic sinistral movements between Iberia and Europe have been postulated in several plate kinematic reconstructions since the 70's. The North Pyrenean Fault has been indicated as the Iberia-Europe Mesozoic plate boundary that should have accommodated such a huge amount of strike-slip/transtensive motion. As reported by the authors, there are currently no firm geological constraints supporting significant sinistral deformation during the Jurassic or the Cretaceous along this fault. The authors thus propose that the Mesozoic strike-slip movement could have partly occurred along the Ebro-W Iberia boundary. In detail, they propose that along this boundary, the Asturian, Maestrat, Cameros, and Columbretes basins formed/were reactivated as pull apart basins within a lithospheric Mesozoic sinistral strike-slip shear zone, where hundreds of km of sinistral motion would have occurred. The authors do not individuate and describe the lithospheric fault(s) bordering the pull apart system and ensuring the connection of the sinistral shear zone with the Bay of Biscay and the north Atlantic. As far I know, the only candidate is the 400 km long Ventaniella Fault. Thus, it is mandatory to describe and discuss the nature and kinematics of this fault. Apart from this, my impression is that using Ventaniella + North Pyrenean faults instead of the North Pyrenean fault along, is jumping out of the frying pan into the fire: The Ventaniella fault is well exposed and only gently affected by Cenozoic deformation. Paleozoic markers across it are presently offsetted in a dextral sense of less than 5 km (see Alvarez-Marrón, 1995. Journal of Structural Geology or any published geological map of the Cantabrian region). The dextral movement for the Ventaniella fault is generally attributed to a Cenozoic stage. One may argue that the amount of this Cenozoic displacement could be not well constrained (Mesozoic sinistral + cenozoic dextral). However, you can use the Cenozoic dextral displacement of the 100 km long Ubierna fault, which significantly overlaps the Ventaniella fault at its SE tip, to get an idea of the order of magnitude. For the Ubierna fault, the Cenozoic dextral displacement proposed by different authors ranges from 10 km (see Tavani et al., 2011, Tectonophysics) to almost nothing (see Quintana et al., 2015, Tectonophysics). Thus, if we remove 0 to 10 km of Cenozoic dextral displacement for the Ventaniella fault, we

end up with Paleozoic markers displaced in a sinistral sense - during the Mesozoic - of less than 5 km. This issue should be addressed.

Minor points Line 2 well registered L3 a key L4 The Late Permian-Triassic Iberian rift basins have accommodated... L8 reconstruction, we L19-21 and orogens. However, the required.....often uncertain. L 42 Understood by who? Also, here and below it must be clearly differentiated between papers in which the strike-slip motion is postulated/suggested, from those in which evidence of strike-slip tectonics is documented L 46 list the evidence L59 I suggest to briefly mention the permo-triassic stratigraphy of the area. L71. Remove pre-salt (no salt has been introduced to the reader) L76-81. Poorly relevant L82-83. Rephrase it L90-3. Cryptic L93-94. Expand the concept. L 94-104. This is material for the discussion. L96 breakup ( L112-115. Add Alvaro et al '79. Alvaro, M., del Villar, R. C., & Vegas, R. (1979). Un modelo de evolución geotectónica para la Cadena Celtibérica L119. See Gomez et al 2002 for a partial subsidence curve in the Basque-Cantabrian basin. Additional curves can be probably derived from papers published in the book "The Geology of Spain". L139. As it stands, it seems that Rat and Aurell have suggested left-lateral tectonics, which is not the case. L174. Label them in figure 4

Figs 2&3. Increase the font size

---

## Referee Comment (RC2) · Alexander L. Peace (Referee) · 23 Mar 2020

This short paper by Angrand et al. makes some interesting and relevant points regarding the evolution of Iberia. The description of geological events that shaped the region is detailed and well organised, needing only minor modifications and clarifications in my opinion. The subject of the paper is very timely and is suitable for Solid Earth. However, I felt that the paper required further work to be suitable for publication. In particular, the description of the methodology, the quality of the figures and some other aspects outlined below need improving. Thus, my overall recommendation is revision of the manuscript as it think it has the potential to make a good contribution to Solid

[Figure]

Earth.

1) Description of reconstruction methodologies, workflow and examination of previous reconstructions

The paper essentially revolves around detailed examination of plate reconstructions to explore specific aspects of Iberia's evolution. This is a combination of previous reconstructions and the authors own work. This is a worthy topic for investigation given that Iberia's kinematics are a source of substantial unknows when conducting reconstructions of this region.

As such, given that the paper is based on plate reconstructions, my main issue with the paper is that the methods related to plate reconstructions are not currently well described. This is in part because the methods are merged in with the description of the regional evolution. In addition, it was not immediately clear which aspects of the reconstructions are the authors own work and what is from previous reconstructions. I would therefore suggest separating out the workflow and methods into a dedicated section.

I also felt that because multiple reconstructions are referred to further examination of the limitations and inputs of these models is required. For example, many reconstructions have been produced for the region recently (Müller et al., 2016; Barnett-Moore et al., 2018; Nirrengarten et al., 2018; Peace et al., 2019a). Each of these models comes with simplifications and limitations depending on the aspects examined (e.g., local/global models and rigid/deformable models) and I felt that this needed further examining in the manuscript. In addition, given that the rotations for different parts of the model presented are from different previous work I felt that a summary of the poles used would be highly beneficial. This could be simply achieved in a summary table showing pole timing and location with the corresponding reference. Table 1 currently does not adequately display the required information and although the 'motion paths' on Figures 4 and 5 help somewhat they are quite hard to read.

[Figure]

**2) Kinematics of minor plates**

Related to the previous point, the concussion of the paper that previous work has neglected the need for a Ebro microcontinent/plate/block seems reasonable and adds of a growing bank of work demonstrating that such smaller blocks play a crucial role in such rift systems. Separating Iberia into smaller 'plates' seems reasonable given the information presented. However, it is apparent that even within relatively coherent plates/blocks there is some deformation but at what point is such an entity an independent plate? This is particularly pertinent as the boundaries between the plates are described in the manuscript 'diffuse'. The nature of diffuse deformation has been the focus of recent deformable modelling of the region which might be of use to the authors (Peace et al., 2019a). One of the problems encountered in Peace et al. (2019a) is the over thickening of crust related to strike slip deformation. Perhaps the authors could shed some insights here.

Also, I felt that description of how the kinematics of such blocks are defined requires further clarification and description. By this I mean that the large-scale kinematics of the major plates can be reconstructed from the oceanic isochrons for the Mesozoic, but this is not the case for the minor plates. The minor plates instead rely on much poorer constraints, such as timing of syn-rift sedimentation and faulting styles (as used by the authors). As such, I felt that further information on how the kinematics of Iberia's constituent plates were reconstructed is required. In addition, I felt that this aspect could have been better reconciled with the geological observations. This point may in part be rectified by addressing the point above regarding the methods.

One of the main conclusions of the work presented in the manuscript is that breaking Iberia into smaller blocks in plate tectonic models might result in more realistic reconstructions (i.e. emphasis on the Ebro block). This is in line with a number of recent studies in the region that also use smaller blocks (e.g., Nirrengarten et al., 2018). Thus, I think it should be more clearly outlined that the conclusion of the present paper supports those of the previous work. Moreover, breaking plates into smaller plates/blocks

with independent kinematics presents several issues that need considering further. For example, the requirement of substantial amounts of strike slip deformation for the authors model would benefit from further examination of the geological evidence. I acknowledge that this is examined by in the manuscript somewhat but I think it could be clearer.

My final point regarding minor plates is that the authors focus on minor plates in Iberia appears to not extend to the other parts of the modelled region which I think likely over simplifies the region and perhaps the interpretation. This is demonstrated in Figures 4 and 5 where the separate Ebro and West Iberia blocks are clearly visible but not the separate blocks included in the recent models such as the Flemish Cap, Orphan Knoll, Porcupine Bank etc. (e.g., Nirrengarten et al., 2018). The importance of including these blocks is shown in Peace and Welford (2020). Essentially, these blocks play an important kinematic role and I do not think that Iberia can be accurately reconstructed without including these blocks. I suggest that the authors try to include these blocks or discuss why they are not included.

Other minor points

Although the paper is generally relatively easy to follow there are a number of sections and sentences that require further work and clarification. Many of these are outlined in the minor points below but I suggest the authors also give the text a thorough proof read prior to resubmission. In addition, I found that although the figures convene much of the relevant information to support the paper I felt that they needed substantial work to be of publication quality. In particular, the text and other details need to be increased in size substantially. Specific aspects of the figures that need addressing are outlined in the minor points below.

Line 3 – 'rift systems'. Consider adding 'spreading' to this sentence as breakup has actually occurred in the region.

line 4 – 'significant'. Is it possible to quantify how significant?

Introduction – The opening paragraph has no citations despite containing several statements that require citations. I suggest adding relevant citations to the opening paragraph.

Line 17 – 'plate tectonic reconstructions'. As you have shown not all reconstructions are necessarily based on oceanic magnetic isochrons. I think this should be clarified.

Line 20 – I suggest adding relevant citations after 'boundaries'.

Lines 27-29 - This sentence doesn't make complete sense to me. Perhaps 'if' should be replaced with 'although'?

Lines 29-30 – This sentence is confusing. I suggest rewording.

Line 46 – 'evidence'. What sort of evidence. I suggest providing further details of this 'evidence'.

Section 2 – I found this whole section quite wordy and hard to follow. I suggest refining it down to just the most essential details.

Line 60 – 'Iberian Buffer'. If this is a quote perhaps it should have a reference?

Line 66 – 'Atlantic province and Northwest Europe'. I feel like these locations and citations could be better organised. I suggest separating out the regions better and adding the citations that are appropriate for the specific region. Also see Sandoval et al. (2019) and Yang et al. (2020) for very recent southern North Atlantic margins work.

Line 83 – 'abnormally high heat flow'. Abnormally high compared to what value? What is normal heat flow anyway?

Line 87 – I feel that this sentence overly simplifies the relationship between CAMP and the breakup. I suggest seeing Peace et al. (2019b) for a detailed review of this.

Lines 94-95 – I found this sentence quite awkward to follow and suggest rewording it.

Line 96 – A space is missing before the citation.

Line 99 – 'the' is possibly missing before 'Pangea'?

Line 99-100 – A review of insulation beneath Pangea is undertaken in Peace et al. (2019b).

Lines 110-115 – I felt that this paragraph would benefit from several references.

Lines 121-122 – Are you talking about Beta factor or stretching factor here? Please clarify.

Line 153 – Were the same blocks used here as those from Nirrengarten et al. (2018) and subsequently Peace et al. (2019a)? Or are they different? I suggest clarifying either way.

Line 155 – I suggest expanding upon why a 'full fit' reconstruction of Iberia is not possible? I suspect that some of the troubles are stemming from the inclusion on the Flemish Cap as part of the North American plate rather than an independent plate. Also a brief discussion of breakup anomalies offshore Iberia might be useful here.

Line 165 – 'workflow'. I think a dedicated workflow section would be beneficial.

Line 220 – I think it would be useful to summarise the rotations described in the text as a table.

Line 239 – Why does the Iberia-Ebro boundary have a more complex tectonic history than the Europe-Ebro boundary? I suggest explaining this further.

Line 256 – Awkward wording. I suggest editing this phrase.

Figure 1 – The text is too small to read on the geological time scale.

Figures 2-3 – Details and text are too small to read on all parts of these figures. I would also suggest more clearly labelling the subfigures and describing them more fully in the captions.

References

Barnett-Moore, N., Müller, R.D., Williams, S., Skogseid, J., and Seton, M., 2018, A reconstruction of the North Atlantic since the earliest Jurassic: Basin Research, v. 30, p. 160–185, doi:10.1111/bre.12214.

Müller, R.D. et al., 2016, Ocean Basin Evolution and Global-Scale Plate Reorganization Events Since Pangea Breakup: Annual Review of Earth and Planetary Sciences, v. 44, p. 107–138, doi:10.1146/annurev-earth-060115-012211.

Nirrengarten, M., Manatschal, G., Tugend, J., Kusznir, N., and Sauter, D., 2018, Kinematic evolution of the southern North Atlantic: implications for the formation of hyper-extended rift systems: Tectonics, p. 2, doi:10.1002/2017TC004495.

Peace, A.L., Welford, J.K., Ball, P.J., and Nirrengarten, M., 2019a, Deformable plate tectonic models of the southern North Atlantic: Journal of Geodynamics, doi:10.1016/j.jog.2019.05.005.

Peace, A., Phethean, J., Franke, D., Foulger, G.R., Schiffer, C., Welford, J.K., McHone, G., Rocchi, S., Schnabel, M., and Doré, A.., 2019b, A review of Pangaea dispersal and Large Igneous Provinces – In search of a causative mechanism: Earth-Science Reviews, doi:10.1016/j.earscirev.2019.102902.

Peace, A.L., and Welford, J.K., 2020, "Conjugate margins?"–An oversimplification of the complex southern North Atlantic rift and spreading system? Interpretation, v. 8, p. 1–54. https://doi.org/10.1190/int-2019-0087.1

Sandoval, L., Welford, J.K., MacMahon, H., and Peace, A.L., 2019, Determining continuous basins across conjugate margins: The East Orphan, Porcupine, and Galicia Interior basins of the southern North Atlantic Ocean: Marine and Petroleum Geology, v. 110, p. 138–161, doi:10.1016/j.marpetgeo.2019.06.047.

Yang, P., Welford, J.K., Peace, A.L., and Hobbs, R., 2020, Tectonophysics Investigating the Goban Spur rifted continental margin, offshore Ireland, through integration of new seismic reflection and potential field data: Tectonophysics, v. 777,

doi:10.1016/j.tecto.2020.228364.

---

## Author Comment (AC1) · 14 Apr 2020

1. Response to the major point

We acknowledge the comment done by an anonymous referee about our manuscript. This comment raises a major issue about a geological implication of our model that needs to be clarified. This is about the role of the Ventaniella Fault as a good candidate to accommodate the Iberia-Ebro relative motion.

This point is critical as most of the recent models accounting for a segmented Iberian plate invoke the Ventaniella Fault as a main boundary fault accommodating a large

amount of Iberia-Ebro movement (Tugend et al., 2015; Nirrengarten et al., 2018). As pointed out by the referee, the estimated left-lateral displacement along the Ventaniella Fault (after correction of the right-lateral Cenozoic displacement) is expected to be in the order a some kilometers during late Paleozoic-middle Cretaceous (Tavani et al., 2011).

We actually do not propose that the Ventaniella Fault accommodates in our reconstruction the large displacement between the Ebro block and Europe. We made this point clearer by reorganizing the discussion (please note that the shape of the final version in the revised manuscript may differ a little bit from this one). The structure of the discussion is now:

"Discussion: Implications for strike-slip movements and the Europe-Iberia boundary

1. Amount of strike-slip displacement # former discussion section

2. Strike-slip structures in the intra-Iberian basins # new section

Despite the requirement of 245 km left-lateral strike-slip displacement along the Iberia-Ebro boundary from 160 to 100 Ma, there are no simple geological evidence in support of a unique major crustal-scale fault in the Iberian Range-Basque Cantabrian Basin system.

Several studies have suggested that a left-lateral shear zone can be recognized along the Iberian Range and the Basque-Cantabrian rifts system. Geological evidence includes the High Tagus Fault in the Iberian Range (Aldega et al., 2019; Aurell et al., 2019) and the Ventaniella Fault in the Basque-Cantabrian region (e.g., Tavani et al., 2011). The latter fault is often considered in recent reconstructions to accommodate alone the Iberia-Ebro movement (Tugend et al., 2015; Nirrengarten et al., 2018). However, the estimated left-lateral displacement along the Ventaniella Fault is only in the order of magnitude of a few kilometers (Tavani et al., 2011) and therefore cannot be used as a North Pyrenean Fault equivalent.

[Figure]

In the Basque Cantabrian Basin, the Ventaniella Fault is part of a NW-SE fault system that acted as left-lateral shear zone during the Late Jurassic-Early Cretaceous and has been subsequently inverted with a right-lateral kinematic during the Cenozoic (De Vicente et al., 2011; Tavani et al., 2011; Cámara, 2017). These faults have a Triassic origin (Tavani and Granado, 2015). Tectonic activity along these faults gets younger NE-ward (Ubierna fault: Late Jurassic-Early Cretaceous; Zamanza-Oña fault: Early-Middle Cretaceous; salt tectonics in the center of the basin, Cámara, 2017). The required Iberia-Ebro displacement could have been distributed along these structures.

The role of the weak Triassic evaporites in efficiently decoupling deformation in the presalt basement from the thin-skinned extension in sedimentary cover has been emphasized largely in the Pyrenees (Lagabrielle et al., 2020; Grool et al., 2019; Duretz et al., 2019; Jourdon et al., 2020). Salt tectonics has also been suggested to have been particularly significant from the Jurassic through the Early Cretaceous in Mesozoic basins that shaped NW-directed boundary between Ebro and Iberia, including the Basque-Cantabrian Basin (Cámara, 2017), Parentis Basin (Ferrer et al., 2012), Cameros Basin (Rat et al., 2019) and Maestrat Basin (Vergés et al., 2020). The surface expression of the crustal strike-slip movements is inferred to have been limited in supra-salt layers."

2. Responses to minor points

L2: done

L3: done

L4: we shortened this sentence: "The Late Permian-Triassic Iberian rift basins have accommodated extension, but. . .Ăă"

L8: done

L19-21: done

L42: "An alternative scenario has recently emerged (Tugend et al., 2015; Nirrengarten et al., 2018, Tavani et al., 2018), proposing a spatiotemporal partitioning of the deformation in a wider deformation corridor than the single Pyrenean belt. It suggests that the major strike-slip movement required to accommodate the eastwards movement of Iberia first occurred during the Late Jurassic-Early Cretaceous in Northern Iberia along the NW-SE-trending Iberian Massifs. Indeed, along these massifs, several extensional basins registered major subsidence and strike-slip deformation during the Late Permian to middle Cretaceous time interval (Álvaro et al., 1978; Salas and Casas, 1993; Salas et al., 2001; Aldega et al., 2019; Aurell et al., 2019, Soto et al., 2019)."

L 46: merged with L42.

L 59: We add a sentence about the stratigraphy. "This late Permian-Lower Triassic phase is associated with the deposition of thick detrital non-marine deposits in intra-continental basins. Sedimentation became carbonaceous during the middle Triassic. Finally, the Late Triassic is characterized by a thick evaporitic (mainly salt) sequence (e.g., Orti et al., 2017)."

L 71: We keep this as we added informations about the stratigraphy.

L 76-81: This paragraph is needed to introduce the following paragraph.

L 82-83: We reorganized this paragraph according to RC2's comments.

L 90-93 and L 93-94: "The persistence of shallow-marine to non-marine deposition during this period contrasts with the large accommodation space that is required at larger scale to sediment the giant evaporitic province in the Late Permian (Jackson et al., 2019) and in the Late Triassic (Stolfova and Shannon, 2009; Leleu et al., 2016; Orti et al., 2017). Therefore subsidence associated with crustal thinning expected for this period does not fit with increase of tectonic subsidence as predicted by modeling (McKenzie, 1978)."

L 94-104: We prefer keeping this paragraph in that position as it supports the extension phase we discuss later in the text.

L 96: done

L112-115: done

L119: We agree that it would be interesting to add more subsidence data but we chose on purpose to limit the number of curves to not overload the figure. In addition, and as reported by the referee, subsidence curves in Gomez et al. (2002) are not covering the all Mesozoic and do not show the Triassic phase.

L 139: we removed "strike slip deformation".

L 174: done.

3. References

Aldega, L., Viola, G., Casas-Sainz, A., Marcén, M., Román-Berdiel, T., and van der Lelij, R.: Unraveling Multiple Thermotectonic Events Accommodated by Crustal-Scale Faults in Northern Iberia, Spain: Insights From K-Ar Dating of Clay Gouges, Tectonics, 38, 3629–3651, https://doi.org/10.1029/2019TC005585, 2019.

Alvaro, M. (1979). Modelo De Evolución Geotectónica Para. Acta Geològica His-pànica, 14, 172–177.

Aurell, M., Fregenal-Martínez, M., Bádenas, B., Muñoz-García, M. B., Élez, J., Meléndez, N., and de Santisteban, C.: Middle Jurassic–Early Cretaceous tectono-sedimentary evolution of the southwestern Iberian Basin (central Spain): Major palaeo-geographical changes in the geotectonic framework of the Western Tethys, Earth-Science Reviews, 199, 102 983, https://doi.org/10.1016/j.earscirev.2019.102983, https://doi.org/10.1016/j.earscirev.2019.102983, 2019.

Cámara, P. (2017). Salt and Strike-Slip Tectonics as Main Drivers in the Structural Evolution of the Basque-Cantabrian Basin, Spain. In Permo-Triassic Salt Provinces of Europe, North Africa and the Atlantic Margins (pp. 371-393). Elsevier.

De Vicente, G., Cloetingh, S. A. P. L., Van Wees, J. D., & Cunha, P. P. (2011). Tectonic classification of Cenozoic Iberian foreland basins. Tectonophysics, 502(1-2), 38-61.

Duretz, T., Asti, R., Lagabrielle, Y., Brun, J. P., Jourdon, A., Clerc, C., and Corre, B.: Numerical modelling of Cretaceous Pyrenean Rifting: The interaction between mantle exhumation and syn-rift salt tectonics, Basin Research, pp. 1–16, https://doi.org/10.1111/bre.12389, 2019.

Ferrer, O., Jackson, M. P. A., Roca, E., & Rubinat, M. (2012). Evolution of salt structures during extension and inversion of the Offshore Parentis Basin (Eastern Bay of Biscay). Geological Society, London, Special Publications, 363(1), 361-380.

Gómez, M., Vergés, J., & Riaza, C. (2002). Inversion tectonics of the northern margin of the Basque Cantabrian Basin. Bulletin de La Societe Geologique de France, 173(5), 449–459. https://doi.org/10.2113/173.5.449

Grool, A. R., Huismans, R. S., and Ford, M.: Salt décollement and rift inheritance controls on crustal deformation in orogens, Terra Nova, 31, 562–568, https://doi.org/10.1111/ter.12428, 2019.

Jackson, C. A. L., Gawthorpe, R. L., Elliott, G. M., & Rogers, E. R. (2019). Salt thickness and composition influence rift structural style , northern North Sea , offshore Norway. July 2018, 514–538. https://doi.org/10.1111/bre.12332

Jourdon, A., Mouthereau, F., Pourhiet, L. L., and Callot, J. P.: Topographic and Tectonic Evolution of Mountain Belts Controlled by Salt Thickness and Rift Architecture, pp. 1–14, https://doi.org/10.1029/2019TC005903, 2020.

Lagabrielle, Y., Asti, R., Duretz, T., Clerc, C., Fourcade, S., Teixell, A., Labaume, P., Corre, B., and Saspiturry, N.: A review of cretaceous smooth-slopes extensional basins along the Iberia-Eurasia plate boundary: How pre-rift salt controls the modes of continental rifting and mantle exhumation, Earth-Science Reviews, 201, 103 071, https://doi.org/10.1016/j.earscirev.2019.103071, https://doi.org/10.1016/j.earscirev.2019.103071, 2020.

Leleu, S., Hartley, A. J., van Oosterhout, C., Kennan, L., Ruckwied, K., & Gerdes,

K. (2016). Structural, stratigraphic and sedimentological characterisation of a wide rift system: The Triassic rift system of the Central Atlantic Domain. Earth-Science Reviews, 158, 89–124. https://doi.org/10.1016/j.earscirev.2016.03.008

McKenzie, D. (1978). Some remarks on the development of sedimentary basins. Earth and Planetary Science Letters, 40(1), 25–32. https://doi.org/10.1016/0012-821X(78)90071-7

Nirrengarten, M., Manatschal, G., Tugend, J., Kusznir, N. J., and Sauter, D.: Kinematic Evolution of the Southern North Atlantic: Implications for the Formation of Hyperextended Rift Systems, Tectonics, 37, 89–118, https://doi.org/10.1002/2017TC004495, 2018.

Ortí, F., Pérez-López, A., & Salvany, J. M. (2017). Triassic evaporites of Iberia: Sedimentological and palaeogeographical implications for the western Neotethys evolution during the Middle Triassic–Earliest Jurassic. Palaeogeography, Palaeoclimatology, Palaeoecology, 471, 157–180. https://doi.org/10.1016/j.palaeo.2017.01.025

Rat, J., Mouthereau, F., Brichau, S., Crémades, A., Bernet, M., Balvay, M., Ganne, J., Lahfid, A., and Gautheron, C.: Tectonothermal Evo- lution of the Cameros Basin: Implications for Tectonics of North Iberia, Tectonics, 38, 440–469, https://doi.org/10.1029/2018TC005294, 2019.

Salas, R., & Casas, A. (1993). Mesozoic extensional tectonics, stratigraphy and crustal evolution during the Alpine cycle of the eastern Iberian basin. Tectonophysics, 228(1–2), 33–55. https://doi.org/10.1016/0040-1951(93)90213-4

Salas, R., Guimerà, J., Mas, R., Martín-Closas, C., Melendez, A., & Alonso, A. (2001). Evolution of the Mesozoic Central Iberian Rift System and its Cainozoic inversion (Iberian chain). In Mémoires du Muséum national d'histoire naturelle (Vol. 186, Issue 2).

Soto, R., Casas-Sainz, A. M., Oliva-Urcia, B., García-Lasanta, C., Izquierdo-Llavall,

E., Moussaid, B., Kullberg, J. C., Román-Berdiel, T., Sánchez-Moya, Y., Sopeña, A., Torres-López, S., Villalaín, J. J., El-Ouardi, H., Gil-Peña, I., & Hirt, A. M. (2019). Triassic stretching directions in Iberia and North Africa inferred from magnetic fabrics. Terra Nova, 31(5), 465–478. https://doi.org/10.1111/ter.12416

Štolfová, K., & Shannon, P. M. (2009). Permo‐Triassic development from Ireland to Norway: basin architecture and regional controls. Geological Journal, 44(6), 652–676.

Tugend, J., Manatschal, G., and Kusznir, N. J.: Spatial and temporal evolution of hyper-extended rift systems: Implication for the nature, kinematics, and timing of the Iberian-European plate boundary, Geology, 43, 15–18, https://doi.org/10.1130/G36072.1, 2015.

Tavani, S., Bertok, C., Granado, P., Piana, F., Salas, R., Vigna, B., & Muñoz, J. A. (2018). The Iberia-Eurasia plate boundary east of the Pyrenees. Earth-Science Reviews, 187(August), 314–337. https://doi.org/10.1016/j.earscirev.2018.10.008

Tavani, S., & Granado, P. (2015). Along‐strike evolution of folding, stretching and breaching of supra‐salt strata in the Plataforma Burgalesa extensional forced fold system (northern Spain). Basin Research, 27(4), 573-585.

Tavani, S., Quintà, A., & Granado, P. (2011). Cenozoic right-lateral wrench tectonics in the Western Pyrenees (Spain): the Ubierna Fault System. Tectonophysics, 509(3-4), 238-253.

Vergés, J., Poprawski, Y., Almar, Y., Drzewiecki, P.A., Moragas, M., Bover-Arnal, T., Macchiavelli, C., Wright, W., Messager, G., Embry, J.-C., Hunt, D., 2020. Tectono-Sedimentary Evolution of Jurassic-Cretaceous diapiric structures: Miravete anticline, Maestrat Basin, Spain. Basin Research n/a. doi:10.1111/bre.12447

---

## Author Comment (AC2) · 4 May 2020

Responses to the major points

We thank Alexander L. Peace for his is a very constructive review that led us to improve the manuscript, in particular the presentation of the methodology. Find below our responses to the major points raised by the reviewer and further minor points.

1. Description of the reconstruction method

We add a dedicated 'Methodology' section before the section 'Kinematics of Iberia between Atlantic and Tethys'. In this new section we present the published kinematic

models used in our reconstruction and the modifications we made. We added a new figure to present the models from the literature and a new table to present the rotation poles of the main plates of our GPlates model.

**2. Kinematics of minor plates**

**a. Definition of an independent plate**

A very interesting point raised in this comment is about the definition of what is an individual plate. We define Ebro as a continental block rather than an independent plate. We think that this definition is better appropriate than 'plate', as its motion cannot be simply related to the forces typically driving lithospheric plates such as mantle convection, slab pull, ridge push etc. No localized plate boundaries can be defined such as spreading centers or subduction zones. Rather, the Ebro block represents a rigid continental body surrounded by deformed areas moving independently between 'plates'. Further study is required to fully understand the origin, nature and evolution of the Ebro block and we cannot answer to these questions in the present manuscript.

In the light of recent publications (e.g., Nirrengarten et al., 2018; Peace et al., 2019a), which show the importance of intra-plate deformation in plate kinematic models, the definition of what is a tectonic plate needs to be thought not only in term of continental region bounded by oceanic crust. In our manuscript, we have adopted the terminology "block" to Ebro whereas Europe, Africa and Western Iberia, all bordered by spreading centers, are plates.

**b. Strike-slip deformation**

Concerning the crustal thickening related to the strike-slip deformation, we also experienced difficulties to accurately represent the strike-slip deformed areas using the GPlates topological network over such large distances (∼200 km). This results in inappropriate mesh and local high strain (both compressive or extensive). The precise study of these deformed areas is out of the scope of this paper. However, this is something we are working on. We would be very interested to discuss this further in order to establish a proper methodology that would apply to strike-slip settings.

c. Breaking Iberia into small blocks

Recent works that separate the Ebro continental block from Iberia (Tugend et al., 2015; Nirrengarten et al., 2018) are presented in the Introduction. We add a sentence in the discussion to better say that the conclusion about breaking Iberia into smaller blocks supports previous works. According to Referee #1 comments, we also better discuss the nature of the Iberia-Ebro tectonic boundary. 'Our results support recent studies (e.g., Tugend et al., 2015; Nirrengarten et al., 2018) that postulate that breaking Iberia into smaller blocks results in more realistic models.'

d. N Atlantic blocks

The N Atlantic blocks (Flemish Cap, Orphan Knoll, Porcupine) are included in our GPlates model. These blocks follow the kinematics of Peace et al. (2019) although they are not distinctly represented (they are delimited by the background diffuse area and the fault features), nor studied with the same intention, in our study. In the revised manuscript we make an effort to better represent these blocks in the figures but we do not want to overload the figure too much.

Responses to the minor points

L 3: we replaced 'rift systems' by 'oceanic systems'

L 4: we shortened this sentence: "The Late Permian-Triassic Iberian rift basins have accommodated extension, but. . .Ăǎ"

Introduction: "Global plate tectonic reconstructions are mostly based on the knowledge and reliability of magnetic anomalies that record age, rate and direction of sea- floor spreading (Stampfli and Borel, 2002; Müller et al., 2008; Seton et al., 2012). Where these constraints are lacking or their recognition ambiguous, kinematic reconstructions rely on the description and interpretation of the structural, sedimentary, igneous and

metamorphic rocks of rifted margins and orogens (e.g., Handy et al., 2010; McQuarrie and Van Hinsergen, 2013). However, the required quantification and distribution of finite strain into deformed continents remain often uncertain due to the poor preservation of pre-kinematic markers."

L 17: cf Introduction.

L 20: cf Introduction.

L27-29: done

L 29-30: "Because of the lack of geological constraints about the timing and localization, this displacement has been supposedly exported along the North Pyrenean Fault."

L 46: Reworked based on Referee #1's comments.

Section 2: we reworked this section based on both referees' comments.

L 60: This is not a quote.

L 66: Now reads as follows: 'Crustal thinning, attested by thick late Permian-Triassic detrital rift-basins deposited above an erosive surface, is well documented on seismic lines along the Atlantic margins (Fig. 2): Nova Scotia-Moroccan basins (Welsink et al., 1989; Deptuck & Kendell, 2017; Hafid, 2000); Iberia-Grand Banks (Balkwill & Legall, 1989; Leleu et al., 2016; Spooner et al., 2018); southern North Atlantic (Tankard & Welsink, 1987; Doré, 1991, Doré et al., 1999; Štolfová & Shannon, 2009; Peace et al., 2019a; Sandoval et al., 2020); North Western Approaches (Avedik, 1975; Evans et al., 1990; McKie, 2017; North Sea, McKie, 2017; Jackson et al., 2018; Hassan et al., 2019; Phillips et al., 2019).

Onshore Iberia (Arche & López-Gómez, 1996; Soto et al., 2019) and in the Pyrenean-Provence domains (Lucas, 1985; Espurt et al., 2019; Cámara & Flinch, 2017; Bestani et al., 2016) (Fig. 1b), an angular unconformity is observed between the Paleozoic and

the Permian-Triassic strata (Fig. 3).'

L 83; L 87: Now reads as follows: 'An expression of the continued lithospheric thinning and thermal instability associated with high heat flow during the Permian (McKenzie et al., 2015) and the Triassic (Peace et al., 2019b, and references therein). Lithospheric extension prior (or associated with the premises of the subsequent) Early Jurassic continental breakup in the Central Atlantic then favored drainage of mantle melt reservoir (Silver et al., 2006; Peace et al., 2019b), attested by the very rapid emergence of the widespread tholeiitic magmatic CAMP (Central Atlantic Magmatic Province) event at the Triassic-Jurassic boundary (∼200 Ma) in the Central Atlantic (Olsen, 1997; Marzoli et al., 1999; McHone, 2000). The CAMP extends to Iberia as large-scale volcanic intrusions such as the Messejana-Plasencia dyke (Cebriá et al., 2003) in Iberia and the Late Triassic-Early Jurassic ophitic magmatism in the Pyrenees (e.g., Azambre et al., 1987). Extension and salt movements in the North Sea basins during the Late Triassic further point to the propagation of the North Atlantic rift (Goldsmith et al., 2003).'

L 94-95: Now reads as follows: 'Two hypotheses may be invoked to explain the difference with the McKenzie model. (1) Reduction of mantle density during lithospheric thinning, due to mantle phase transitions to lighter mineral phases because of crustal attenuation (Simon and Podladchikov, 2008) and/or due to the trapping of melt in the rising asthenosphere before breakup (Quirk and Rüpke, 2018) in addition to magmatic re-thickening of attenuated crust by underplating. (2) Another possible hypothesis for the Permian-Triassic topographic evolution. . .'

L 96: done

L 99: done

L 99-100: Now reads as follows: 'Another possible hypothesis for the Permian-Triassic topographic evolution of the Iberian basins relies on the complex post-Variscan evolution of the Iberian lithosphere. Recent studies have shown that during the existence of Pangea supercontinent (∼300 to ∼200 Ma), temperature in the asthenospheric man-

tle increased due to the thermal insulation by the continental lid (Coltice et al., 2009; Ganne et al., 2016). This thermal insulation would be responsible for the accumulation of magmatic material of the CAMP (see Peace et al. 2019b, and references therein). Such mantle thermal anomaly could have further inhibited lithospheric mantle re-equilibration after late-Variscan mantle delamination over a long-time span. This model requires a strong impermeability of the overlying lithosphere (Silver et al., 2006). Once mantle temperature dropped as a consequence of the Pangea breakup and magmatic emission at the Triassic/Jurassic boundary, lithospheric mantle started to cool and thicken, causing isostatic subsidence of the thinned Iberian crust and resulting in topographic drop.'

L 110-115: This part is mostly a description of our figures. We however added some references.

L 121-122: We calculated the stretching factor (so called beta factor) from from the tectonic subsidence, as defined in Watts (2001).

L 153; L 155; L 165; L 220: done with the Methodology section.

L 239: done

L 256: Now reads as follows: 'To revolve several long-lasting problems of the Mesozoic kinematics of Iberia, we propose to better consider: the late Permian-Triassic basins evolution in Iberian kinematic reconstructions, the role of the Ebro continental block in the partitionning of the deformation, and to replace Iberia in a larger-scale plate reconstruction of the Atlantic and Tethys domains. We show that: (1) left-lateral strike-slip movement did occur in the Pyrenees from the late Permian to the Early Cretaceous but ended as the Bay of Biscay opened, (2) late Permian-Triassic extension in the Atlantic and Iberia (including Ebro) is key to quantify the strike-slip movement in Iberia that is otherwise not well resolved from the geological constraints in Iberian basins and from full-fit reconstructions in the Jurassic. Salt tectonics that decouples syn-rift Iberian basins evolution from their basement likely explains the lack of geological constraints.'

Figures: We increased font size and better described the figures and subfigures in the captions.

---

## Author Response (AR1)

Paul Angrand, PhD
GET-OMP CNRS UMR5563
16 Av. Edouard Belin
31400 Toulouse
5 France
Corresponding author: Paul Angrand (paul.angrand@get.omp.eu)

May 12, 2020

Dear Editor, dear Reviewers,

Thank you very much for the time you have dedicated to review and comment our manuscript. We believe that your com-
10 ments have helped us to improve significantly the quality of the work. Please find below the responses to the Referees.

The response is organized with responses to the Referees' major points, structured as follow: (1) comments from Referees
(italicized text), (2) authors' response, (3) authors' changes in manuscript, and responses to the Referees' minor points. For
minor changes (e.g., spelling, word substitution), we indicate if we applied the changes but not systematically write the modified
text in the response (if we simply apply, we indicate by "done").

15 In addition, please find a marked-up manuscript version showing the changes made (red, small size = removed, blue =
modifications; made with latexdiff in LaTeX, as suggested).

Significant changes have been made to the manuscript. Main changes include a new dedicated section about the methodology
we use for the reconstruction and a subsection in the discussion to better discuss geological implications of our reconstruction.

20 Yours sincerely,
Paul Angrand
On the behalf of co-authors Frédéric Mouthereau, Emmanuel Masini and Riccardo Asti.

**1 Response to interactive comment of Referee #1 (Anonymous)**

**1.1 Response to the major point**

25 **General comment** *This is a short paper dealing with the long lasting problem of the Mesozoic kinematics of Iberia. Here the authors revise the the Permo-Triassic rifting stage in Iberia and surrounding regions, and propose that including this stage into the puzzle may help in reconciling geological evidence and plate kinematic models. In detail, the authors suggest that Iberia cannot be considered an integer plate but, rather, it must be separated into the Ebro and Western Iberia blocks, which is in agreement with most of the recently published works on this topic/area. The work is well written and well illustrated. There are*

30 *some minor points that should be addressed and a major issue.*

**Authors' response** We acknowledge the constructive review done by an anonymous referee about our manuscript. This review helped us to improve our manuscript, by better discussing the geological implications of our reconstruction.
* * *
**Referee's comment** *Hundreds of km of Mesozoic sinistral movements between Iberia and Europe have been postulated in several plate kinematic reconstructions since the 70's. The North Pyrenean Fault has been indicated as the Iberia-Europe*

35 *Mesozoic plate boundary that should have accommodated such a huge amount of strike- slip/transtensive motion. As reported by the authors, there are currently no firm geological constraints supporting significant sinistral deformation during the Jurassic or the Cretaceous along this fault. The authors thus propose that the Mesozoic strike-slip movement could have partly occurred along the Ebro-W Iberia boundary. In detail, they propose that along this boundary, the Asturian, Maestrat, Cameros, and Columbretes basins formed/were reactivated as pull apart basins within a lithospheric Mesozoic sinistral strike-*

40 *slip shear zone, where hundreds of km of sinistral motion would have occurred. The authors do not individuate and describe the lithospheric fault(s) border- ing the pull apart system and ensuring the connection of the sinistral shear zone with the Bay of Biscay and the north Atlantic. As far I know, the only candidate is the 400 km long Ventaniella Fault. Thus, it is mandatory to describe and discuss the nature and kinematics of this fault. Apart from this, my impression is that using Ventaniella + North Pyrenean faults instead of the North Pyrenean fault along, is jumping out of the frying pan into the fire: The Ventaniella fault is*

45 *well exposed and only gently affected by Cenozoic deformation. Paleozoic markers across it are presently offsetted in a dextral sense of less than 5 km (see Alvarez-Marro´n, 1995. Journal of Structural Geology or any published geological map of the Cantabrian region). The dextral movement for the Ventaniella fault is generally attributed to a Cenozoic stage. One may argue that the amount of this Cenozoic displacement could be not well constrained (Mesozoic sinistral + cenozoic dextral). However, you can use the Cenozoic dextral displacement of the 100 km long Ubierna fault, which significantly overlaps the Ventaniella*

50 *fault at its SE tip, to get an idea of the order of magnitude. For the Ubierna fault, the Cenozoic dextral displacement proposed by different authors ranges from 10 km (see Tavani et al., 2011, Tectonophysics) to almost nothing (see Quintana et al., 2015, Tectonophysics). Thus, if we remove 0 to 10 km of Cenozoic dextral displacement for the Ventaniella fault, we end up with Paleozoic markers displaced in a sinistral sense - during the Mesozoic - of less than 5 km. This issue should be addressed.*

**Authors' response** This comment raises a major issue about a geological implication of our model that needs to be clarified.

55 This is about the role of the Ventaniella Fault as a good candidate to accommodate the Iberia-Ebro relative motion.

This point is critical as most of the recent models accounting for a segmented Iberian plate invoke the Ventaniella Fault as a main boundary fault accommodating a large amount of Iberia-Ebro movement (Tugend et al., 2015; Nirrengarten et al., 2018). As pointed out by the referee, the estimated left-lateral displacement along the Ventaniella Fault (after correction of the right-lateral Cenozoic displacement) is expected to be in the order a some kilometers during late Paleozoic-middle Cretaceous

60 (Tavani et al., 2011).

We actually do not propose that the Ventaniella Fault accommodates in our reconstruction the large displacement between the Ebro block and Europe. We made this point clearer by reorganizing the discussion.

**Changes in the manuscript**

"Discussion: Implications for strike-slip movements and the Europe-Iberia boundary 1. Amount of strike-slip displacement

65 # former discussion section

2. Strike-slip structures in the intra-Iberian basins # new section

Despite the requirement of 245 km left-lateral strike-slip displacement along the Iberia-Ebro boundary from 160 to 100 Ma, there is no simple geological evidence in support of a unique major crustal-scale fault in the Iberian Range-Basque Cantabrian Basin system.

Several studies have suggested that a left-lateral shear zone can be recognized along the Iberian Range and the Basque-Cantabrian rifts system. Geological evidence includes the High Tagus Fault in the Iberian Range (Aldega et al., 2019; Aurell et al., 2019) and the Ventaniella Fault in the Basque-Cantabrian region (e.g., Tavani et al., 2011). The latter fault is often considered in recent reconstructions to accommodate alone the Iberia-Ebro movement (Tugend et al., 2015; Nirrengarten et al., 2018). However, the estimated left-lateral displacement along the Ventaniella Fault is only in the order of magnitude of a few kilometers (Tavani et al., 2011) and therefore cannot be used as a North Pyrenean Fault equivalent.

In the Basque Cantabrian Basin, the Ventaniella Fault is part of a NW-SE fault system that acted as left-lateral shear zone during the Late Jurassic-Early Cretaceous and has been subsequently inverted with a right-lateral kinematic during the Cenozoic (De Vicente et al., 2011; Tavani et al., 2011; Cámara and Flinch, 2017). These faults have a Triassic origin (Tavani and Granado, 2015). Tectonic activity along these faults gets younger NE-ward (Ubierna fault: Late Jurassic-Early Cretaceous; Zamanza-Oña fault: Early-Middle Cretaceous; salt tectonics in the center of the basin, Cámara and Flinch, 2017). We suggest the Iberia-Ebro displacement have possibly been distributed along these structures.

The role of the weak Triassic evaporites in efficiently decoupling deformation in the pre-salt basement from the thin-skinned extension in sedimentary cover has been emphasized largely in the Pyrenees (e.g., Grool et al., 2019; Duretz et al., 2019; Jourdon et al., 2020; Lagabrielle et al., 2020). Salt tectonics has also been suggested to have been particularly significant from the Jurassic through the Early Cretaceous in Mesozoic basins that shaped NW-directed boundary between Ebro and Iberia, including the Basque-Cantabrian Basin (Cámara and Flinch, 2017), Parentis Basin (Ferrer et al., 2012), Cameros Basin (Rat et al., 2019) and Maestrat Basin (Vergés et al., 2020). The surface expression of the crustal strike-slip movements is inferred to have been limited in supra-salt layers."

**1.2 Responses to minor points**

Comments from Referees are in italicized text.

L2: *well registered* > "well recorded"

L3: *a key* > done

L4: *The Late Permian-Triassic Iberian rift basins have accommodated. . .* > we shortened this sentence: « The Late Permian-Triassic Iberian rift basins have accommodated extension, but. . . »

L8: *reconstruction. We* > done

L19-21: *and orogens. However, the required.....often uncertain.* > "and orogens (e.g., Handy et al., 2010). However, ..."

L42: *Understood by who? Also, here and below it must be clearly differentiated between papers in which the strike-slip motion is postulated/suggested, from those in which evidence of strike-slip tectonics is documented* > We reworked this part in order to better describe the previous models. Now reads as follow: "An alternative scenario has recently emerged (Tugend et al., 2015; Nirrengarten et al., 2018; Tavani et al., 2018), proposing a spatiotemporal partitioning of the deformation in a wider deformation corridor than the single Pyrenean belt. It suggests that the transcurrent deformation that results from the eastwards movement of Iberia occurred mainly during the Late Jurassic-Early Cretaceous in Northern Iberia along a hundred-kilometer scale pull-part or en-echelon rift basins formed by the NW-SE-trending Iberian Massifs. Indeed, along these massifs, several extensional basins recorded major subsidence and strike-slip deformation during the Late Permian to middle Cretaceous time interval (Alvaro et al., 1979; Salas et al., 2001; Aldega et al., 2019; Aurell et al., 2019; Soto et al., 2019). However, no geological evidence for lithosphere-scale strike-slip movements is yet clearly defined in the intra-Iberian basins."

L 46: *list the evidence* > merged with L42.

L 59: *I suggest to briefly mention the permo-triassic stratigraphy of the area.* > We add a sentence about the stratigraphy. "This late Permian-Lower Triassic phase is associated with the deposition of thick detrital non-marine deposits in intra-continental basins. Sedimentation became carbonaceous during the middle Triassic. Finally, the Late Triassic is characterized by a thick evaporitic (mainly salt) sequence (e.g., Ortí et al., 2017)."

L 71: *Remove pre-salt (no salt has been introduced to the reader)* > We keep this as we added informations about the stratigraphy.

L 76-81: *Poorly relevant* > This paragraph is needed to introduce the following paragraph.

L 82-83: *Rephrase it* > We reorganized this paragraph according to RC2's comments.

L 90-93: *Cryptic* and L 93-94:*Expand the concept* > Now reads as follow: "The persistence of shallow-marine to non-marine deposition during this period contrasts with the large accommodation space that is required at larger scale to sediment the giant evaporitic province in the Late Permian (Jackson et al., 2019) and in the Late Triassic (Štolfová and Shannon, 2009; Leleu et al., 2016; Ortí et al., 2017). Therefore the subsidence appears much lower than that predicted by simple isostatic model of crustal thinning (McKenzie, 1978)."

L 94-104: *This is material for the discussion* > We prefer keeping this paragraph in that position as it supports the extension phase we discuss later in the text.

L 96: *breakup (* > done

L112-115: *Add Alvaro et al '79* > done

L119: *See Gomez et al 2002 for a partial subsidence curve in the Basque-Cantabrian basin. Additional curves can be probably derived from papers published in the book "The Geology of Spain".* > We agree that it would be interesting to add more subsidence data but we chose on purpose to limit the number of curves to not overload the figure. Most importantly, and as reported by the referee, subsidence curves in the proposed source are not covering the all Mesozoic and do not show the Triassic phase.

L 139: *As it stands, it seems that Rat and Aurell have suggested left-lateral tectonics, which is not the case* > we removed « strike slip deformation ».

L 174: *Label them in figure 4* > done

Figs 2&3. *Increase the font size* > done.

**2 Response to interactive comment of Referee #2 (A. L. Peace)**

**2.1 Responses to the major points**

**General comment** *This short paper by Angrand et al. makes some interesting and relevant points regard- ing the evolution of Iberia. The description of geological events that shaped the region is detailed and well organised, needing only minor modifications and clarifications in my opinion. The subject of the paper is very timely and is suitable for Solid Earth. However, I felt that the paper required further work to be suitable for publication. In particular, the description of the methodology, the quality of the figures and some other aspects outlined below need improving. Thus, my overall recommendation is revision of the manuscript as it think it has the potential to make a good contribution to Solid Earth.*

**Authors' response** We thank Alexander L. Peace for his is a very constructive review that led us to improve the manuscript, in particular the presentation of the methodology. Find below our responses to the major points raised by the reviewer and further minor points.
* * *
**1) Description of reconstruction methodologies, workflow and examination of previous reconstructions.** *The paper essentially revolves around detailed examination of plate reconstructions to explore specific aspects of Iberia's evolution. This is a combination of previous reconstructions and the authors own work. This is a worthy topic for investigation given that Iberia's kinematics are a source of substantial unknows when conducting reconstructions of this region.*

*As such, given that the paper is based on plate reconstructions, my main issue with the paper is that the methods related to plate reconstructions are not currently well described. This is in part because the methods are merged in with the description of the regional evolution. In addition, it was not immediately clear which aspects of the reconstructions are the authors own work and what is from previous reconstructions. I would therefore suggest separating out the workflow and methods into a dedicated section.*

*I also felt that because multiple reconstructions are referred to further examination of the limitations and inputs of these models is required. For example, many reconstructions have been produced for the region recently (Mu¨ller et al., 2016; Barnett-Moore et al., 2018; Nirrengarten et al., 2018; Peace et al., 2019a). Each of these models comes with simplifications and limitations depending on the aspects examined (e.g., local/global models and rigid/deformable models) and I felt that this needed further examining in the manuscript. In addition, given that the rotations for different parts of the model presented are from different previous work I felt that a summary of the poles used would be highly beneficial. This could be simply achieved in a summary table showing pole timing and location with the corresponding reference. Table 1 currently does not adequately display the required information and although the 'motion paths' on Figures 4 and 5 help somewhat they are quite hard to read.*

**Authors' response Description of the reconstruction method** We add a dedicated 'Methodology' section before the section 'Kinematics of Iberia between Atlantic and Tethys'. In this new section we present the published kinematic models used in our reconstruction and the modifications we made. We added a new figure to present the models from the literature and a new table to present the rotation poles of the main plates of our GPlates model.

**Changes in the manuscript**

"4 Methodology

4.1 Previous kinematic models

[revised manuscript text omitted]

Because a full-fit reconstruction in the Southwest Iberia leads to significant overlapping between the Flemish Cap and Galicia, we use the Nazaré Fault (Pereira et al., 2017) to segment West Iberia. This allows us to minimize the overlap of Northwest Iberia (Galicia) over Flemish Cap, or to have a gap between Southwest Iberia and Newfoundland."
* * *
**2) Kinematics of minor plates.** *Related to the previous point, the concussion of the paper that previous work has neglected the need for a Ebro microcontinent/plate/block seems reasonable and adds of a growing bank of work demonstrating that such smaller blocks play a crucial role in such rift systems. Separating Iberia into smaller 'plates' seems reasonable given the information presented. However, it is apparent that even within relatively coherent plates/blocks there is some deformation but at what point is such an entity an independent plate? This is particularly pertinent as the boundaries between the plates are described in the manuscript 'diffuse'. The nature of diffuse deformation has been the focus of recent deformable modelling of the region which might be of use to the authors (Peace et al., 2019a). One of the problems encountered in Peace et al. (2019a) is the over thickening of crust related to strike slip deformation. Perhaps the authors could shed some insights here.*

*Also, I felt that description of how the kinematics of such blocks are defined requires further clarification and description.*

*By this I mean that the large-scale kinematics of the major plates can be reconstructed from the oceanic isochrons for the Mesozoic, but this is not the case for the minor plates. The minor plates instead rely on much poorer constraints, such as timing of syn-rift sedimentation and faulting styles (as used by the authors). As such, I felt that further information on how the kinematics of Iberia's constituent plates were reconstructed is required. In addition, I felt that this aspect could have been better reconciled with the geological observations. This point may in part be rectified by addressing the point above regarding the methods.*

*One of the main conclusions of the work presented in the manuscript is that breaking Iberia into smaller blocks in plate tectonic models might result in more realistic re- constructions (i.e. emphasis on the Ebro block). This is in line with a number of recent studies in the region that also use smaller blocks (e.g., Nirrengarten et al., 2018). Thus, I think it should be more clearly outlined that the conclusion of the present paper sup- ports those of the previous work. Moreover, breaking plates into smaller plates/blocks with independent kinematics presents several issues that need considering further. For example, the requirement of substantial amounts of strike slip deformation for the authors model would benefit from further examination of the geological evidence. I acknowledge that this is examined by in the manuscript somewhat but I think it could be clearer.*

*My final point regarding minor plates is that the authors focus on minor plates in Iberia appears to not extend to the other parts of the modelled region which I think likely over simplifies the region and perhaps the interpretation. This is demonstrated in Figures 4 and 5 where the separate Ebro and West Iberia blocks are clearly visible but not the separate blocks included in the recent models such as the Flemish Cap, Orphan Knoll, Porcupine Bank etc. (e.g., Nirrengarten et al., 2018). The importance of including these blocks is shown in Peace and Welford (2020). Essentially, these blocks play an important kinematic role and I do not think that Iberia can be accurately reconstructed without including these blocks. I suggest that the authors try to include these blocks or discuss why they are not included.*

**Authors' responses**

**Definition of an independent plate** A very interesting point raised in this comment is about the definition of what is an individual plate. We define Ebro as a continental block rather than an independent plate. We think that this definition is better appropriate than 'plate', as its motion cannot be simply related to the forces typically driving lithospheric plates such as mantle convection, slab pull, ridge push etc. No localized plate boundaries can be defined such as spreading centers or subduction zones. Rather, the Ebro block represents a rigid continental body surrounded by deformed areas moving independently between 'plates'. Further study is required to fully understand the origin, nature and evolution of the Ebro block and we cannot answer to these questions in the present manuscript.

In the light of recent publications (e.g., Nirrengarten et al., 2018; Peace et al., 2019b), which show the importance of intra-plate deformation in plate kinematic models, the definition of what is a tectonic plate needs to be thought not only in term of continental region bounded by oceanic crust. In our manuscript, we have adopted the terminology "block" to Ebro whereas Europe, Africa and Western Iberia, all bordered by spreading centers, are plates.

**Strike-slip deformation** Concerning the crustal thickening related to the strike-slip deformation, we also experienced difficulties to accurately represent the strike-slip deformed areas using the GPlates topological network over such large distances ( 200 km). This results in inappropriate mesh and local high strain (both compressive or extensive). The precise study of these deformed areas is out of the scope of this paper. However, this is something we are working on. We would be very interested to discuss this further in order to establish a proper methodology that would apply to strike-slip settings.

**Breaking Iberia into small blocks** Recent works that separate the Ebro continental block from Iberia (Tugend et al., 2015; Nirrengarten et al., 2018) are presented in the Introduction. We add a sentence in the discussion to better say that the conclusion about breaking Iberia into smaller blocks supports previous works. According to Referee #1 comments, we also better discuss the nature of the Iberia-Ebro tectonic boundary.

**Changes in the manuscript**

'Our results support recent studies (e.g., Tugend et al., 2015; Nirrengarten et al., 2018) that postulate that breaking Iberia into smaller blocks results in more realistic models.'

**N Atlantic blocks** The N Atlantic blocks (Flemish Cap, Orphan Knoll, Porcupine) are included in our GPlates model. These blocks follow the kinematics of Peace et al. (2019b) although they are not distinctly represented (they are delimited by the background diffuse area and the fault features), nor studied with the same intention, in our study. In the revised manuscript we make an effort to better represent these blocks in the figures but we do not want to overload the figure too much.

**2.2 Responses to the minor points**

Comments from Referees are in italicized text.

L 3: *'rift systems'. Consider adding 'spreading' to this sentence as breakup has actually occurred in the region* > We replaced 'rift systems' by 'oceanic systems'

280 L 4: *'significant'. Is it possible to quantify how significant?* > we shortened this sentence: « The Late Permian-Triassic Iberian rift basins have accommodated extension, but... »

Introduction: *The opening paragraph has no citations despite containing several state- ments that require citations. I suggest adding relevant citations to the opening para- graph.* > We added references. Now reads as follow: "Global plate tectonic reconstructions are mostly based on the knowledge and reliability of magnetic anomalies that record age, rate and direction of

285 sea-floor spreading (Stampfli and Borel, 2002; Müller et al., 2008; Seton et al., 2012). Where these constraints are lacking or their recognition ambiguous, kinematic reconstructions rely on the description and interpretation of the structural, sedimentary, igneous and metamorphic rocks of rifted margins and orogens (e.g., Handy et al., 2010; McQuarrie and Van Hinsbergen, 2013). However, the required quantification and distribution of finite strain into deformed continents remain often uncertain due to the poor preservation of pre-kinematic markers."

290 L 17: *'plate tectonic reconstructions'. As you have shown not all reconstructions are necessarily based on oceanic magnetic isochrons. I think this should be clarified.* > cf Introduction.

L 20: *I suggest adding relevant citations after 'boundaries'.* > cf Introduction.

L27-29: *This sentence doesn't make complete sense to me. Perhaps 'if' should be replaced with 'although'?* > done

L 29-30: *This sentence is confusing. I suggest rewording.* > Now reads as follow: 'Because of the lack of geological con-

295 straints about the timing and localization, this displacement has been supposedly exported along the North Pyrenean Fault.'

L 46: *'evidence'. What sort of evidence. I suggest providing further details of this 'evidence'.* > Reworked based on Referee #1's comments.

Section 2: *I found this whole section quite wordy and hard to follow. I suggest refining it down to just the most essential details.* > we reworked this section based on both referees' comments.

300 L 60: *'Iberian Buffer'. If this is a quote perhaps it should have a reference?* > This is not a quote. This not a standard way to name Iberia, so we prefer to use quote marks here.

L 66: *'Atlantic province and Northwest Europe'. I feel like these locations and citations could be better organised. I suggest separating out the regions better and adding the citations that are appropriate for the specific region. Also see Sandoval et al. (2019) and Yang et al. (2020) for very recent southern North Atlantic margins work.* > We reworked this part and included more

305 references. Now reads as follows: "Crustal thinning, attested by thick late Permian-Triassic detrital rift-basins deposited above an erosive surface, is well documented on seismic lines along the Atlantic margins (Fig. 2): Nova Scotia-Moroccan basins (Welsink et al., 1989; Deptuck and Kendell, 2017; Hafid, 2000); Iberia-Grand Banks (Balkwill and Legall, 1989; Leleu et al., 2016; Spooner et al., 2019); southern North Atlantic (Tankard and Welsink, 1987; Doré, 1991; Doré et al., 1999; Štolfová and Shannon, 2009; Peace et al., 2019b; Sandoval et al., 2019); North Western Approaches (Avedik, 1975; Evans, 1990; McKie,

310 2017); North Sea (McKie, 2017; Jackson et al., 2019; Hassaan et al., 2020; Phillips et al., 2019). Onshore Iberia (Arche and López Gómez, 1996; Soto et al., 2019) and in the Pyrenean-Provence domains (Lucas, 1985; Espurt et al., 2019; Cámara and Flinch, 2017; Bestani et al., 2016) (Fig. 1b), an angular unconformity is observed between the Paleozoic and the Permian-Triassic strata (Fig. 2)."

L 83; L 87: *'abnormally high heat flow'. Abnormally high compared to what value? What is normal heat flow anyway?* AND

315 *I feel that this sentence overly simplifies the relationship between CAMP and the breakup. I suggest seeing Peace et al. (2019b) for a detailed review of this.* > We reworked this part, as we agree that it was unclear. As suggested by the referee, it is difficult to define a 'abnormally high heat flow'. Now reads as follows: "An expression of the continued lithospheric thinning and thermal instability associated with high heat flow during the Permian (McKenzie et al., 2015) and the Triassic (Peace et al., 2019a, and references therein). Lithospheric extension prior (or associated with the premises of the subsequent) Early Jurassic continental

320 breakup in the Central Atlantic then favored drainage of mantle melt reservoir (Silver et al., 2006; Peace et al., 2019a), attested by the very rapid emergence of the widespread tholeiitic magmatic CAMP (Central Atlantic Magmatic Province) event at the Triassic-Jurassic boundary ( 200 Ma) in the Central Atlantic (Olsen, 1997; Marzoli et al., 1999; McHone, 2000). The CAMP extends to Iberia as large-scale volcanic intrusions such as the Messejana-Plasencia dyke (Cerbiá et al., 2003) in Iberia and the

Late Triassic-Early Jurassic ophitic magmatism in the Pyrenees (e.g., Azambre et al., 1987). Extension and salt movements in the North Sea basins during the Late Triassic further point to the propagation of the North Atlantic rift (Goldsmith et al., 2003)."

L 94-95: *I found this sentence quite awkward to follow and suggest rewording it.* > Now reads as follows: "Two hypotheses may be invoked to explain the difference with the McKenzie model. (1) Reduction of mantle density during lithospheric thinning, due to mantle phase transitions to lighter mineral phases because of crustal attenuation (Simon and Podladchikov, 2008) and/or due to the trapping of melt in the rising asthenosphere before breakup (Quirk and Rüpke, 2018) in addition to magmatic re-thickening of attenuated crust by underplating. (2) Another possible hypothesis for the Permian-Triassic topographic evolution..."

L 96: *A space is missing before the citation.* > done

L 99: *'the' is possibly missing before 'Pangea'?* > done

L 99-100: *A review of insulation beneath Pangea is undertaken in Peace et al. (2019b).* > We modified this section to better consider the review in (Peace et al., 2019a). Now reads as follows: "Another possible hypothesis for the Permian-Triassic topographic evolution of the Iberian basins relies on the complex post-Variscan evolution of the Iberian lithosphere. Recent studies have shown that during the existence of Pangea supercontinent ( 300 to 200 Ma), temperature in the asthenospheric mantle increased due to the thermal insulation by the continental lid (Coltice et al., 2009; Ganne et al., 2016). This thermal insulation would be responsible for the accumulation of magmatic material of the CAMP (see Peace et al., 2019a, and references therein). Such mantle thermal anomaly could have further inhibited lithospheric mantle re-equilibration after late-Variscan mantle delamination over a long-time span. This model requires a strong impermeability of the overlying lithosphere (Silver et al., 2006). As a consequence of the Pangea breakup and magmatic emission at the Triassic/Jurassic boundary, lithospheric mantle started to cool and thicken, causing isostatic subsidence of the thinned Iberian crust and resulting in topographic drop."

L 110-115: *I felt that this paragraph would benefit from several references.* > This part is mostly a description of our figures. We however added some references.

L 121-122: *Are you talking about Beta factor or stretching factor here? Please clarify.* > We calculated the stretching factor (so called beta factor) from from the tectonic subsidence, as defined in Watts (2001).

L 153: *Were the same blocks used here as those from Nirrengarten et al. (2018) and subsequently Peace et al. (2019a)? Or are they different? I suggest clarifying either way.* > cf Methodology section

L 155: *I suggest expanding upon why a 'full fit' reconstruction of Iberia is not possible? I suspect that some of the troubles are stemming from the inclusion on the Flemish Cap as part of the North American plate rather than an independent plate. Also a brief discussion of breakup anomalies offshore Iberia might be useful here.* > cf Methodology section

L 165: *'workflow'. I think a dedicated workflow section would be beneficial.* > cf Methodology section

L 220: *I think it would be useful to summarise the rotations described in the text as a table.* > cf Methodology section. We added a new table with rotation poles of the main plates.

L 239: *Why does the Iberia-Ebro boundary have a more complex tectonic history than the Europe-Ebro boundary? I suggest explaining this further.* > "More complex" was ambiguous we changes with: "The Iberia-Ebro boundary has played as right-lateral and left-lateral kinematics."

L 256: *Awkward wording. I suggest editing this phrase.* > Now reads as follows: "To revolve several long-lasting problems of the Mesozoic kinematics of Iberia, we propose to better consider: the late Permian-Triassic basins evolution in Iberian kinematic reconstructions, the role of the Ebro continental block in the partitionning of the deformation, and to replace Iberia in a larger-scale plate reconstruction of the Atlantic and Tethys domains. We show that: (1) left-lateral strike-slip movement did occur in the Pyrenees from the late Permian to the Early Cretaceous but ended as the Bay of Biscay opened, (2) late Permian-Triassic extension in the Atlantic and Iberia (including Ebro) is key to quantify the strike-slip movement in Iberia that is otherwise not well resolved from the geological constraints in Iberian basins and from full-fit reconstructions in the Jurassic. Salt tectonics that decouples syn-rift Iberian basins evolution from their basement likely explains the lack of geological constraints."

Figures: *The text is too small to read on the geological time scale.* AND *Details and text are too small to read on all parts of these figures. I would also suggest more clearly labelling the subfigures and describing them more fully in the captions.* > We increased font size and generally better described the figures and subfigures in the captions.

[revised manuscript text omitted]

Crustal thinning

Crustal thinning, attested by thick late Permian-Triassic detrital rift-basins deposited above an erosive surface, is well documented in the Atlantic province and Northwest Europe continental shelves on seismic lines along the Atlantic margins (Fig. 1b)(Ziegler et al., 2004; Ziegler and Dèzes, 2006; Leleu et al., 2016; Müller et al., 2016; Spooner et al., 2019; Soto et al., 2019), by thick late Permian-Triassic detrital basins of the North Western Approaches (Avedik, 1975; Evans, 1990; McKie, 2017), North Atlantic (Tankard and Welsink, 1987; Doré, 1991; Doré et al., 1999; Štolfová and Shannon, 2009; Peace et al., 2019b) and North Sea (McKie, 2017; Jackson et al., 2019; Hassaan et al., 2020; Phillips et al., 2019) (Fig. 2). The late Permian-Early Triassic pre-salt extension is well imaged on seismic lines 2): Nova Scotia-Moroccan basins (Welsink et al., 1989; Deptuck and Kendell, 2017; Hafid, 2000); Iberia-Grand Banks (Balkwill and Legall, 1989; Leleu et al., 2016; Spooner et al., 2019); southern North Atlantic (Tankard and Welsink, 1987; Doré, 1991; Doré et al., 1999; Štolfová and Shannon, 2009; Peace et al., 2019b; Sandoval et al., 2019); North Western Approaches (Avedik, 1975; Evans, 1990; McKie, 2017); North Sea (McKie, 2017; Jackson et al., 2019; Hassaan et al., 2020; Phillips et al., 2019). Onshore Iberia (Arche and López Gómez, 1996; Soto et al., 2019) and in the Pyrenean-Provence domains (Lucas, 1985; Espurt et al., 2019; Cámara and Flinch, 2017; Bestani et al., 2016) (Fig. 2). An 1b), an angular unconformity is observed at the base of the late Permian-Early Triassic in the Western Approaches (Evans, 1990), West Iberia (Leleu et al., 2016), Nova Scotia (Welsink et al., 1989; Deptuck and Kendell, 2017), the Grand Banks (Balkwill and Legall, 1989), the Moroccan basins (Hafid, 2000), in the Pyrenees (Lucas, 1985; Espurt et al.,

2019; Cámara and Flinch, 2017; Bestani et al., 2016) and throughout Iberia between the Paleozoic and the Permian-Triassic strata (Arche and López Gómez, 1996). (Fig. 2).

The Permian tectonic phase is contemporaneous with widespread magmatism related to breakup of Pangea, and its transition toward diffuse extension. This is observed in present-day rifted margins of the North Atlantic such as the North Sea and Norwegian-Danish Basins (Glennie et al., 2003), the Western Approach (McKie, 2017), the Scottish Midland Valley (Upton et al., 2004) and in the basement of Cenozoic collision belts around Iberia, for instance, in the Pyrenees (Lago et al., 2004a; Denèle et al., 2012; Vacherat et al., 2017; Saspiturry et al., 2019), Iberian Range (Lago et al., 2004b), Catalan Coastal Ranges (Solé et al., 2002), and in the Betic Cordillera (Sánchez-Navas et al., 2017).

An expression of the continued lithospheric thinning and thermal instability associated with high heat flow during the Permian (McKenzie et al., 2015) and the Triassic and abnormally high heat flow is recorded by the (Peace et al., 2019a, and references therein). Lithospheric extension prior (or associated with the premises of the subsequent) Early Jurassic continental breakup in the Central Atlantic then favored drainage of mantle melt reservoir (Silver et al., 2006; Peace et al., 2019a), attested by the very rapid emergence of the widespread tholeiitic magmatic CAMP (Central Atlantic Magmatic Province) event at the Triassic-Jurassic boundary ( 200 Ma) in the Central Atlantic (Olsen, 1997; Marzoli et al., 1999; McHone, 2000). The CAMP extends to Iberia as large-scale volcanic intrusions such as the Messejana-Plasencia dyke (Cerbiá et al., 2003) in Iberia and the Late Triassic-Early Jurassic ophitic magmatism in the Pyrenees (e.g., Azambre et al., 1987). The CAMP may have favored heat dispersion that triggered the subsequent Early Jurassic continental breakup in the Central Atlantic. Extension and salt movements in the North Sea basins during the Late Triassic further point to the propagation of the North Atlantic rift (Goldsmith et al., 2003).

The persistence of shallow-marine to non-marine deposition during this period contrasts with the large accommodation space that is required at larger scale to sediment the giant evaporitic-province in the late evaporitic province in the Late Permian (Jackson et al., 2019) and in the Late Triassic (Štolfová and Shannon, 2009; Leleu et al., 2016; Ortí et al., 2017). Crustal thinning expected for this period therefore does not follow McKenzie 's prediction of subsidence (McKenzie, 1978) 
[revised manuscript text omitted]

**4 Kinematics of Iberia between Atlantic and Tethys Methodology**

A plate reconstruction from late Permian to Cretaceous is presented in Fig. 5 based on a kinematic modelling using GPlates version 2.1 (Müller et al., 2018). This reconstruction aims to present the partitioning of the deformation within Iberia into a larger coherent kinematic model

**4.1 Previous kinematic models**

We compile and implement previous kinematic models involving Iberia (Fig. 4). The objective is to establish a coherent kinematic model of Iberia that considers both the evolution of the Neotethyan and Atlantic regions. These kinematic models are either global (e.g., Müller et al., 2019), based on the assimilation of geological and geophysical information at large scale to allow a dynamic understanding of Earth's plate tectonics but do not aim to solve regional tectonic issues such as strain partitioning between Iberia and Europe. On the other hand, regional models are focused on the reconstruction of North Atlantic (e.g., Barnett-Moore et al., 2016; Nirrengarten et al., 2018; Peace et al., 2019b) or are interested in the reconstruction of the Alpine orogen with inferences on the kinematics of the Tethys and Atlantic Oceans.

555 A critical step in determining the pre-rifting configuration is to restore rifted margins . Here, we adopted the reconstructed continental crust geometry of Nirrengarten et al. (2018) based on a kinematic model of Adria (Schmid et al., 2008; Handy et al., 2010; Van Hinsbergen et al., 2019).

North Atlantic reconstructions use offshore geophysical constraints from the Northwest Iberian margins and pay relatively little attention to the geological evolution of the Pyrenees and other orogenic domains in Iberia (e.g., Sibuet et al.,

195 2004; Barnett-Moore et al., 2016; Nirrengarten et al., 2018). However, these models give fundamental insights on the geometry of the North Atlantic full-fit reconstruction and the timing of oceanic spreading. The nature of some magnetic anomalies in the southern North Atlantic has been the matter of considerable debate (Olivet, 1996; Sibuet et al., 2004; Vissers and Meijer, 2012; Barnett-Moore et al., 2016). Here, we adopt the reconstruction of Nirrengarten et al. (2018) who propose a model based on the re-evaluation of magnetic anomalies that are considered not oceanic before C34 ( 83

200 Ma) and therefore not suitable for kinematic studies (Nirrengarten et al., 2017).

Reconstructions of the Alpine domain (Schmid et al., 2008; Handy et al., 2010) and at a larger scale of the Tethys domain (Van Hinsbergen et al., 2019) are keys to understand the evolution of past oceanic domains now inverted in Alpine orogenic systems (e.g., Paleotethys, Neotethys, Meliata, Pindos and Vardar oceans). These models however do not account for the recent reconstruction of the southern North Atlantic presented above that have impact on the

205 movement of Iberia of interest for our study.

**4.2** Reconstruction workflow

A plate reconstruction from late Permian to middle Cretaceous is presented in Figs. 5 and 6, based on a kinematic modelling using GPlates version 2.1 (Müller et al., 2018). The rotation poles of the main plates are summarized in Table 1 (see also Supplementary Material for GPlates files).

210 Polygons from the model of Seton et al. (2012) were re-defined by including new smaller polygons (continental microblocks) separated by deformed areas in Iberia and Adria to account for internal deformation (Fig.1b).

As Our compilation is as follow: (1) the reconstruction of the western Tethys prior to the Late Jurassic is constrained by the kinematic evolution of the Mediterranean region since the Triassic from Van Hinsbergen et al. (2019) that we corrected for overlap of Iberia over western France; (2) the kinematics of Africa follows Müller et al. (2019), based on Heine et al.

215 (2013); (3) for the Late Jurassic and Cretaceous times, we compiled rotation poles of the North America-Europe system from Barnett-Moore et al. (2016), updated from Peace et al. (2019) for North Atlantic continental blocks (Flemish Cap, Orphan Knoll, and Porcupine Bank); (4) our reconstruction of Adria follows the model from Müller et al. (2019) that was modified to account for the possible younger opening of the Ionian basin (Tugend et al., 2019).

Because there is no motion during the 270-250 Ma interval for the North America and Africa plates relative to Europe

220 (Domeier and Torsvik, 2014), we extended the full-fit cannot be reconstructed along the whole Iberia margin (Fig.

5a), we restore used full-fit only between Northwest Iberia (Galicia) and North America (Flemish Cap) to minimize the strike-slip movement between Iberia and Europe, rather than a full-fit in the Southwest Iberia that leads to significant overlapping between the Flemish Cap and Galicia of these models to 270 Ma.

Our kinematic model is based on These input models were then updated according to the following constraints (Tab. 2): (1) geological constraints on the timing of deformation and subsidence during late Permian-Triassic time in the intra- and peri-Iberian basins mentioned above (Fig. 3); (2) age of rifting,

225 mantle exhumation, onset of oceanic spreading in the Atlantic; (2) the present-day position of ophiolites bodies and the timing of  rifting, oceanic spreading and subduction for the Tethyan-related oceanic domains (Paleotethys, Neotethys, Pindos, Meliata, Vardar); (3) at 100 Ma, Iberia  is close to the present-day  position relative to Europe, so that the  late Mesozoic-Cenozoic Pyrenean shortening is essentially orthogonal.

We then integrate kinematic evolution for published models in both the Atlantic and the Tethys according to the following workflow: 1) the reconstruction of

**4.3 Implementations of the pre-existing models**

230

A critical step in determining the pre-rifting configuration is the restoration of rifted margins. Here, we adopted the reconstructed continental crust geometry of Nirrengarten et al. (2018). Polygons from the model of Nirrengarten et al. (2018) that are based on Seton et al. (2012), were re-defined such that they include new smaller polygons (continental micro-blocks) separated by deformed areas in Iberia and Adria to account for internal deformation (Fig. 1b).

235 The kinematics of these continental blocks (e.g., the Ebro block) has been reconstructed using geological constraints inferred from the western Tethys prior to the Late Jurassic is based on the kinematic evolution of the Mediterranean region since the Triassic from Van Hinsbergen et al. (2019) that we corrected for overlap over the western France, Iberian and Adriatic domains; 2) for the Late Jurassic and Cretaceous times, we compiled rotation poles for Adria and Africa from Handy et al. (2010) and for the North America-Europe system from Barnett-Moore et al. (2016); tectono-sedimentary evolution of intra- and peri-Iberian basins (see Section 3 ) Adria and Africa were then corrected for the position of Africa according to Heine et al. (2013)
[revised manuscript text omitted]